# Routing, Modulation Level, and Spectrum Assignment in Elastic Optical Networks—A Serial Stage Approach with Multiple Sub-Sets of Requests Based on Integer Linear Programming

**Luis Víctor Maidana Benítez [1], Melisa María Rosa Villamayor Paredes [1], José Colbes [1], César F. Bogado-Martínez [1] , Benjamin Barán [2] and Diego P. Pinto-Roa [1,* ]**

[1]  Facultad Politécnica, Universidad Nacional de Asunción, San Lorenzo 2111, Paraguay; lvsmdn@fpuna.edu.py (L.V.M.B.); melisavillamayorp@fpuna.edu.py (M.M.R.V.P.); jcolbes@pol.una.py (J.C.); cfbogado@ing.una.py (C.F.B.-M.)

[2]  School of Informatics, Universidad Comunera, Asunción 1209, Paraguay; bbaran@cba.com.py

[*]  Correspondence: dpinto@pol.una.py

**Abstract:** This paper addresses serialized approaches of the routing, modulation level, and spectrum assignment (RMLSA) problem in elastic optical networks, using multiple sequential sub-sets of requests, under Integer Linear Programming (ILP). The literature has reported two-stage serial optimization methods referred to as RML+SA, which retain computational efficiency when the problem grows, compared to the classical one-stage RMLSA optimization approach. However, there still remain numerous issues in terms of the spectrum used that can be improved when compared to the RMLSA solution. Consequently, this paper proposes RML+SA solutions considering multiple sequential sub-sets of requests, split traffic flow, as well as path-oriented and link-oriented routing models. Simulation results on different test scenarios determine that: (a) the multiple sequential sub-sets of request-based models improve computation time without worsening the spectrum usage when compared to just one set of requests optimization approaches, (b) divisible traffic flow approaches show promise in cases where the number of request sub-sets is low compared to the non-divisible counterpart, and (c) path-oriented routing succeeds in improving the used spectrum by increasing the number of candidate routes compared to link-oriented routing.

**Keywords:** RML+SA; EON; ILP; serial stage; multi-path; multiple sub-sets of requests

## 1. Introduction

Significant technological advances have caused increasing network traffic demand, creating the need for improvements in data transmission [1]. This situation has encouraged the development of Elastic Optical Network (EON) technologies [2]. EON uses a flexible spectrum grid (flex-grid), which has variable spectral width ability per channel according to the demand for bandwidth. Simultaneously, the transponders allow different types of optical signal modulation to achieve efficient spectrum usage, depending on the length of the optical path. Alternatively, EON imposes several restrictions on the optical layer [3] as: (i) continuity, (ii) contiguity, and (iii) non-overlapping of frequency slots (FS) assigned to an optical channel [4]. In this context, the problem of calculating an optical path is called Routing, Modulation Level, and Spectrum Allocation (RMLSA) [5,6], which is an NP-complete problem [7].

The RMLSA problem is enunciated in the following way. Given an EON topology and a set of requests, the RMLSA problem seeks to calculate a physical route, modulation level, and assigned spectrum called light-path to each request minimizing the network optical spectrum usage. The set of light-paths must fulfill the optical layer constraints.

Researchers have developed solutions based on exact and approximate algorithms, leading to a trade-off between optimality and scalability. For example, one-stage methods (typically called RMLSA) based on Integer Linear Programming (ILP) or exhaustive search can calculate optimal solutions at high computational and time costs. In contrast, heuristic and meta-heuristic approaches are scalable solutions but do not guarantee optimality.

Specialized literature reports few efforts for off-line traffic on two-stage serial optimization, called RML+SA [8]. Other strategies that can help improve the RML+SA scalability include: (a) dividing the set of requests into several smaller subsets, and then tackling them consecutively, and (b) splitting the traffic flow into several smaller sub flows. Considering the above strategies, it is necessary to study the impact of the routing scheme based on paths and links for the RML+SA approaches. Consequently, the contributions of this article are the following:

i   Proposal of five new ILP alternatives based on RML+SA strategy, as follows:
    1.  Link-oriented routing, with split traffic flow, and one set of requests.
    2.  Link-oriented routing, with split traffic flow, and multiple sub-sets of requests.
    3.  Link-oriented routing, with unsplit traffic flow, and multiple sub-sets of requests.
    4.  Path-oriented routing, with split traffic flow, and multiple sub-sets of requests.
    5.  Path-oriented routing, with unsplit traffic flow, and multiple sub-sets of requests.

ii  Performance of simulations to study the advantages and limitations of the proposed RML+SA approaches.

The remainder of the present work is organized as follows: Section 2 presents a literature review for the RMLSA problem under off-line traffic. In Section 3, the contributions are presented: strategies and ILP models. Simulation tests are presented in Section 4, while results and discussion are provided in Section 5. Finally, the conclusions and future research lines are given in Section 6.

## 2. Related Works

This section presents a classification of the different published approaches in the specialized literature, considering only point-to-point traffic. The first significant classification of the RMLSA problem considers [9]: on-line, semi-online, and off-line traffic. The on-line RMLSA approaches mainly focus on the problem of spectrum defragmentation and blocking probability [10,11]. Commonly, in these types of scenarios, heuristic algorithms process each request on arrival. This research line is beyond the scope of this work.

On the other hand, it is possible to apply optimization techniques to semi-online and off-line RMLSA problems [12]. In particular, we consider semi-online traffic when requests are met at time intervals [9]. In these groups, we found mathematical programming models and heuristics. To solve the problem, Table 1 shows the main characteristics of those approaches based on ILP and Mixed ILP (MILP). Heuristic approaches are excluded from this study.

Table 1 has twofold sections. The first one reports the state-of-the-art contributions, while the second one reports the models that complement state-of-the-art and are proposals based on the RML+SA of this work. The first five rows consider RMLSA models, while the remaining rows present RML+SA models. The models based on multiple traffic sub-flows are denoted 1LM, MLM, and MPM, while those based on a single traffic flow are denoted ML1 and MP1. The models based on multiple subsets of requests are MLM, ML1, MPM, and MP1. 1LM is the only model proposed with a single set of traffic. Thus, the different configurations of RML+SA-based models are studied in this report.

**Table 1.** RMLSA based on ILP models.

| Optimization Approach | | Request Management | | Routing Strategies | | Traffic Flow Division | | Contributions Reported in the Literature |
|---|---|---|---|---|---|---|---|---|
| One-stage RMLSA | Two-stage RML+SA | (1) One Set of Requests | (M)ultiple Subsets of Requests | (P)ath-oriented routing Pre-calculated Path Table | (L)ink-oriented routing All paths available | (1) One Traffic Flow | (M)ultiple Traffic sub-flows | |
| ✓ | | ✓ | | | ✓ | | ✓ | [13] |
| ✓ | | ✓ | | | ✓ | ✓ | | [13–19] |
| ✓ | | ✓ | | ✓ | | | ✓ | [20] |
| ✓ | | ✓ | | ✓ | | ✓ | | [5,6,17,21–27] |
| ✓ | | | ✓ | ✓ | | ✓ | | [16,28] |
| | ✓ | ✓ | | | ✓ | ✓ | | 1L1 [8] |
| | ✓ | ✓ | | ✓ | | | ✓ | 1PM [29] |
| | ✓ | ✓ | | ✓ | | ✓ | | 1P1 [6] |
| The proposed RML+SA models in this work | | | | | | | | |
| | ✓ | ✓ | | | ✓ | | ✓ | 1LM |
| | ✓ | | ✓ | | ✓ | | ✓ | MLM |
| | ✓ | | ✓ | | ✓ | ✓ | | ML1 |
| | ✓ | | ✓ | ✓ | | | ✓ | MPM |
| | ✓ | | ✓ | ✓ | | ✓ | | MP1 |

As mentioned in the previous section, ILP-based RMLSA approaches are characterized by calculating the optimal solution at a very high computation time cost. The reason for this is that the routing, modulation, and spectrum allocation variables define a considerable large solution space. An alternative is to consider a constructive approach that calculates a sub-optimal solution with low computational consumption, i.e., divide the problem into sub-problems (two stages), solve each sub-problem, and join the sub-solutions. In this context, the RML+SA approach arises in which the RML problem is solved first and then the SA problem [8]. In other words, two solution spaces result much smaller than the one considered by the RMLSA problem. Another aspect that impacts computing time is the set of requests, which can also be divided into subsets, which are resolved consecutively.

Based on the literature review, we can classify the state-of-the-art contributions considering: (1) optimization approach; (2) request management; (3) routing strategy; and (4) allowable traffic flow division. These concepts are explained below:

1. Optimization approaches:
   - One-stage optimization: In this approach, algorithms tackle the RML and SA problems jointly, and it is typically known as the joint RMLSA approach [6,14].
   - Two-stage serial optimization: In this approach, algorithms obtain the solution in two stages; the RML problem is solved first, and then the SA problem is approached as a coloring problem. This approach has been called RML+SA, where the RML phase calculates an ideal cost for the RMLSA problem [6]. On the other hand, it is also possible to calculate the solution in an SA+RML approach [30,31]; however, this approach is outside the scope of this study.

2. Request management:
   - One set of requests: In this approach, which is the most commonly used, the algorithms receive all requests as input to be processed in just one set [6,8,14].
   - Multiple sub-sets of requests: The set of requests is divided into several smaller sub-sets, and then the light-paths are calculated and installed consecutively for each sub-set [28].

3. Routing strategies:
   - Path-oriented routing: Routing algorithms select a path for a request from pre-computed paths [6], typically the $k$ shortest paths.
   - Link-oriented routing: Routing algorithms have all possible routes available [14]; all network links are candidates to be part of a route. No pre-calculated path is necessary.

4. Traffic flow division:
   - One traffic flow: This is the typical routing approach, the whole traffic flow of a request is transmitted on just one lightpath [6,8,14].
   - Multiple traffic sub-flows, also known as multi-path approach [32]: In this situation, the traffic flow can be split into multiple sub-flows and routed independently, therefore, needing a large number of transponders [20].

As shown in Table 1, the field of two-stage serial optimization RML+SA has not been explored enough. Specialized literature reports RML+SA contributions with ILP models that consider the following aspects:

1. One set of Requests, Link-oriented Routing, and One traffic flow, which we name 1L1 [8];
2. One set of Requests, Path-oriented Routing, and Multiple traffic sub-flows (1PM) [29]; and
3. One set of Requests, Path-oriented Routing, and One traffic flow (1P1) [6].

In this context, a study including proposals on ILP models that address these research lines are necessary to determine their benefits and limitations. Consequently, in this work, different ILP-based RML+SA approaches have been developed to cover cases still not considered, as indicated in Table 1 with shaded cells. The study of these cases is

fundamental since the structure of the RML+SA problem influences the type of route and the number of sub-flows. Understanding the benefits and disadvantages of these approaches will help determine whether to apply them in real cases.

As seen in each row of Table 1, we can find several RMLSA contributions that follow the same scheme. In row 2, corresponding to the one-stage, one set of requests, path-oriented routing, and split traffic approach, there are five contributions studying different aspects of the problem [13–19]. In [14], the authors consider the link length to assign the modulation, while in [13] multi-period traffic is studied. In [15], a MILP is proposed for multiple modulation levels but is subject to the spontaneous noise level of the channel. On the other hand, [16] considers linear and nonlinear channel impairments to determine the modulation level. In [17], the authors propose an ILP model that addresses the RMLSA problem for network planning. In [18], the total energy and block consumption are used as the objective function, while in [19], the scheme considers Data Center traffic.

Following the previous analysis, in row 4 of Table 1, we have found several RMLSA approaches oriented to a single set of solutions, path-oriented routing, without traffic flow division [5,6,17,21–26,33]. Notably, [6] proves that the RMLSA is an NP-complete problem. The work [5] differs from [6] in the proposed ILP model and heuristics. Both works have studied the impact of the number of slots on the performance of the networks. In [25], the dynamic and static RMLSA is analyzed using the ILP model and heuristic under the limitations of the optical layer. In particular, they consider variable traffic over time. On the other hand, in [21], energy consumption is considered as an optimization criterion. In [33], an ILP as well as heuristics are proposed that consider a linear combination of network cost, energy consumption, and used spectrum as criteria. In [22], the performance of the Tabu Search in RMLSA is analyzed. In [23], the authors propose an ILP and heuristics for grooming traffic with multiple modulation formats and baud rates for a non-transparent network. In [17], the performances of the ILP-based algorithms and heuristics are analyzed for offline planning, network design, and network reconfiguration problems. An ILP model and heuristics-based adaptive modulation considering spectrum and space allocation are proposed in [26]. In [27], an ILP model and heuristics are proposed to solve the problem of routing, baud rate, coding (code), modulation format, and spectrum allocation.

It is worth mentioning that optimization approaches can also be classified according to the number of objective functions simultaneously optimized [34], considering (a) mono-objective approaches [5,6,8,14–17,19,22,23,26–29,35–37] and (b) multi-objective ones [13,18,20,21,24,25,33,38]. Note that the multi-objective optimization approach, taking into account the Pareto context, was reported for Routing and Spectrum Allocation (RSA) [39,40], but not for RMLSA. In this context, the objective functions considered in the literature propose typically to minimize the maximum slot [5,6,8,16,22,23,28,29]; and in fewer cases minimization of the energy consumption [8,14,20,23], traffic flow [8,14,20,23], cost of used spectrum [22,23,33], the sum of the bit rate of all requests served [17], as well as the difference between requested and assigned spectrum width [20]. Based on the above, this work also considers a mono-objective optimization approach in which we seek to minimize the maximum index of the spectrum.

### 3. Mathematical Programming Models

This section presents the following five proposed RML+SA models based on ILP:

- 1LM = One set of requests, link-oriented routing, and multiple traffic sub-flows.
- MLM = Multiple sub-sets of requests, link-oriented routing, and multiple traffic sub-flows.
- ML1 = Multiple sub-sets of requests, link-oriented routing, and one traffic flow.
- MPM = Multiple sub-sets of requests, path-oriented routing, and multiple traffic sub-flows.
- MP1 = Multiple sub-sets of requests, path-oriented routing, and one traffic flow.

To gain a complete and straightforward perspective of the proposed ILP models, only more complex MLM and MPM models are presented in detail. Subsequently, the

following section explains the parameters that determine the other characteristics of the proposed models 1LM, ML1, and MP1; as well as models already reported in the literature for 1L1 [8], 1PM [29,41], and 1P1 [6]. Particularly, a 1LM approach that is also oriented towards protection has been reported [42]. Protection is also outside the scope of this work. In the specific case of split traffic, we should note that by introducing a continuous variable that determines the traffic rate in each sub-flow, the problem becomes an MILP.

### 3.1. Problem Statement

In EON, the traffic demand between a pair of source and destination nodes is transmitted through multiple sub-carriers or Frequency Slots (FS) or simply –*slots*– for this work. Given a physical topology, a set of requests, and candidate paths, the following conditions need to be met, as follows:

- Satisfy all the source—destination connection demands, determining the route, the modulation format, and the spectrum assignment for each traffic request;
- Optimize spectrum usage minimizing the maximum index of slot used on all optical fibers in the network.

*The following assumptions are established for the proposed models:*

- The spectral bandwidth of each optical fiber is divided into slots.
- The fiber optic capacity in slots terms is equally limited in all links.
- Connection demands are bidirectional, and RMLSA algorithms calculate a light-path for each request.
- Between two light-paths using the same link, there is at least one slot as a guard-band.
- A request is represented by a 3-tuple: $s = (v_o, v_d, \lambda)$, indicating $v_o$ the source node, $v_d$ the destination node and $\lambda$ the requested bandwidth (data rate).
- Multiple sub-sets imply a scheme to approach the problem studied in this work, which does not imply management requirements in the upper layers of the network.
- For the division of traffic flows we consider sliceable bandwidth-variable transponders (SBVTs) mentioned in [13].

Let us consider an example taken from [43], where an EON topology, the modulation format, transmission reach, modulation rate, and the set of requests are given in Figure 1a–d, respectively. This example uses 8 and 16 QAM (Quadrature Amplitude Modulation) as modulation formats, QPSK (Quadrature Phase Shift Keying), and BPSK (Binary Phase Shift Keying) shown in Figure 1b,c with their modulation rate ($R$) and transmission reach ($T$) in Kilometers (Km). The 8QAM is assigned to request $s_1$ because its path length is 500 km, as shown in Figure 1c,d. The 8QAM has a spectral efficiency $R = 3$ bits per symbol; therefore, two slots are assigned to $s_1$, i.e., $\Lambda_1 = \left\lceil \frac{\lambda}{R \cdot C} \right\rceil = \left\lceil \frac{50}{3 \cdot 12.5} \right\rceil = 2$, where the bandwidth of one slot in BPSK is $C = 12.5$ G bps. Similarly, $\Lambda_2 = \left\lceil \frac{35}{4 \cdot 12.5} \right\rceil = 1$ and $\Lambda_3 = \left\lceil \frac{50}{1 \cdot 12.5} \right\rceil = 4$ for $s_2$ and $s_3$, respectively. Figure 1e shows the assigned spectrum, with one slot as guard-band. Note that the spectrum assignment meets the optical layer constraints. The maximum index of slot used in links (or simply –*maximum slot*–) is 9, while the number of slots used in the network is 22. A –*slot block*– is a set of consecutive slots; for example, the consecutive slots 6 to 9 form the slot block of $s_3$.

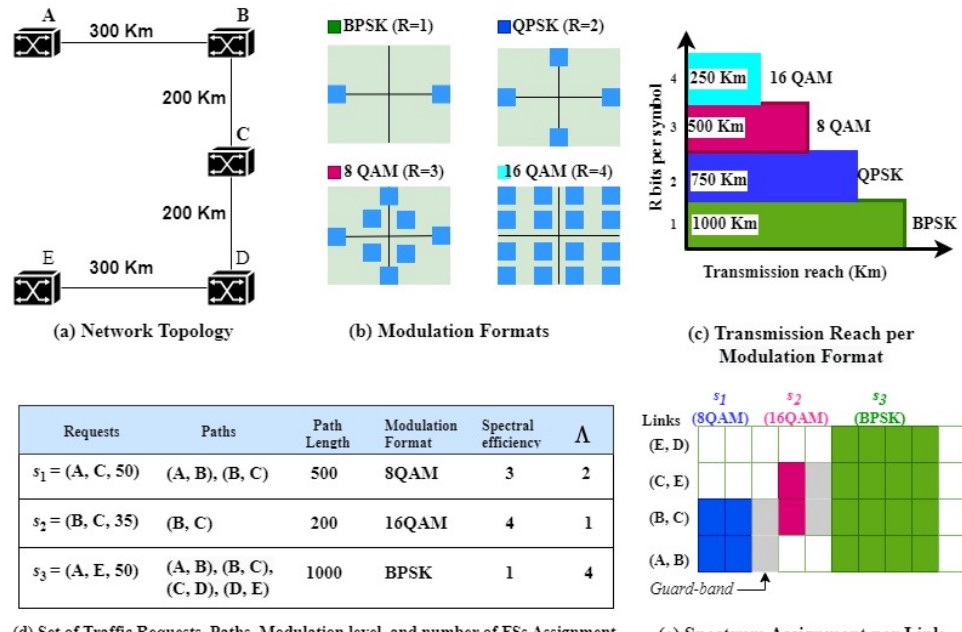

**Figure 1.** Example of an RMLSA Problem [43], (**a**) network topology, (**b**) modulation formats' constellation, (**c**) transmission reach per modulation format type, (**d**) set of traffic requests, modulation assignment and number of slots, and (**e**) spectrum assignment.

### 3.2. Calculation Process for the Multiple Sub-Sets of Requests Approaches

Algorithm 1 presents the general process for the RML+SA approach. For multiple sub-sets of requests-based models, the input set of requests $\mathcal{S} = \{s_1, \ldots, s_{|\mathcal{S}|}\}$ is sorted in descending order considering the request cost (the requested bandwidth $\times$ the shortest path length). Subsequently, the set $\mathcal{S}$ is divided into $RG$ (number of Requests Group) sub-sets or request groups so that $\mathcal{S} = \mathcal{S}_1 \cup \mathcal{S}_2 \cup \cdots \cup \mathcal{S}_{RG}$ and $\mathcal{S}_t \cap \mathcal{S}_{t'} = \varnothing \ (\forall t \neq t')$. In this work, we assume that the sorting and segmentation of $\mathcal{S}$ imply a minimal computational time compared to the optimization process RML+SA.

---

**Algorithm 1** $RMLSA + SA$ for Multiple Sub-sets of Requests Process

---

1: Set up all slots as available $\mathcal{G}_0$;
2: **if** $RG > 1$ **then**
3:     Sort requests of $\mathcal{S}$ in descending order of cost;
4:     Split $\mathcal{S}$ into disjoint sub-sets $\mathcal{S}_1, \ldots, \mathcal{S}_{RG}$;
5: **end if**
6: **for** $t = 1$ **to** $RG$ **do**
7:     Solve RML problem for $\mathcal{S}_t$;
8:     Calculate Matrix $Q$
9:     Solve SA problem for $\mathcal{S}_t$;
10:     Assign calculated light-path to network $\mathcal{G}_t$;
11:     Updated status of slots $\mathcal{G}_t$;
12:     $t \leftarrow t + 1$;
13: **end for**

---

Each $\mathcal{S}_t$ is an input for the RML phase, where remaining groups are processed later (lines 7 to 9 of Algorithm 1). The result of iteration $t$ updates the optical network state, where $\mathcal{V}$ is the set of nodes representing transponders, $\mathcal{L}$ is the set of links representing optical links, and $\mathcal{J}_t$ is the status of slots. One slot can be busy or available to be assigned to requests. This updated state of the graph $\mathcal{G}_t = (\mathcal{V}, \mathcal{L}, \mathcal{J}_t)$ serves as input for the next iteration $t + 1$ since the assigned slots are marked as unavailable resources at subsequent

iterations. In this context, the $\mathcal{G}_t$ represents the graph at the beginning of iteration $t + 1$. Note that in the $\mathcal{G}_0$ all the slots are available for use at iteration 1.

This process is repeated until all $\mathcal{S}_t$ are served. At the end, we have the routes and slots assigned for each request of $\mathcal{S}$. Note that the one set of requests-based approaches are particular cases of the multiple subset of requests, when *RG* = 1, which are also reflected in Algorithm 1. A key aspect is that sub-problems RML and SA are solved consecutively in lines 7 and 9 of Algorithm 1.

After the RML phase, there is an intermediate process before the SA phase starts, where a binary matrix *Q* of dimension $(|\Omega_t| + |\mathcal{S}_t|) \times (|\Omega_t| + |\mathcal{S}_t|)$ is constructed. $\Omega_t$ is the set of requests previously assigned up to the $t - 1$ iteration, i.e., $\Omega_t = \mathcal{S}_1 \cup \ldots \cup \mathcal{S}_{t-1}$ where $\Omega_0 = \varnothing$. *Q* is an adjacency matrix where each matrix entry $Q_{s,s'}$ maps the paths of requests *s* and *s'* previously selected in the RML phase. If $Q_{s,s'}$ = 1, the paths share at least one optical link and 0 otherwise. This matrix defines on which paths to apply the non-overlapping spectrum constraint. At this point, we distinguish two types of requests: the incoming ones $s \in \mathcal{S}_t$ and those already served in the previous steps $\bar{s} \in \Omega_t$.

Due to the structure of the RML+SA approach, the SA phase is the same for both link-oriented and path-oriented routing models.

*3.3. RML Phase Formulation for MPM*

Table 2 presents the notation used in this model. The aim is to calculate an RML solution $X_{sp}$ at the iteration *t* that minimizes the maximum slot used in links $C_{RML}$ subject to a maximum number of sub-flows *K*, considering a sub-set of requests $\mathcal{S}_t$, the state of network resource usage $\mathcal{G}_t$, the sub-sets of already installed requests $\Omega_t$, the set of candidate paths $\mathcal{P}$ and the set of modulation format $\mathcal{M}$. Note that the maximum slot $C_{RML}$ calculated in this phase does not correspond to the maximum slot of the RML+SA problem. It will finally be obtained in the SA phase.

This phase has two steps. The modulation level assignment problem and the routing problem are solved in the first and second step, respectively. The first step assigns the optimal modulation format $M_i \in \mathcal{M}$ and then calculates the number of slots $\Lambda_{sp} = \left\lceil \frac{\lambda^s}{R_i \cdot C} \right\rceil$ to each route $p \in \mathcal{P}_s$. Given a path $p \in \mathcal{P}_s$ and each request, the optimal modulation format $M_i$ is one that uses the maximum modulation rate subject to $LP_s \leq T_i$. Note that the routes of a request may have different modulation formats; therefore, $\Lambda_{sp}$ is input data for the next step.

The second step seeks to determine the *K* routes and the traffic rates carried by these routes for each request. A solution $X_{sp}$ of the routing problem determines the traffic flow ratio for each path of each request. If two solutions have the same $C_{RML}$, the tie is broken by considering the number *B* of used slots in the network. The objective function is a weighted sum between the maximum slot denoted as $C_{RML}$ and the normalized number of slots given by $\frac{B}{|\mathcal{L}|}$ in objective function (1). In short, $\mu$ is chosen in (1) close to 1 to have a lexicographical order with $C_{RML}$ as the most important criterion.

The following formulation expresses the proposed mathematical programming for the RML phase of the MPM model.

$$\text{Minimize } \mu \cdot C_{RML} + (1 - \mu) \cdot \frac{B}{|\mathcal{L}|} \tag{1}$$

subject to:

$$B = \sum_{l \in \mathcal{L}} (F_l - GB). \tag{2}$$

$$\forall l \in \mathcal{L}: \quad C_{RML} \geq F_l - GB. \tag{3}$$

$$\forall l \in \mathcal{L}: \quad F_l = \sum_{s \in \mathcal{S}_t} \sum_{p \in \mathcal{P}_s^l} (\lceil \Lambda_{sp} \cdot X_{sp} \rceil + GB \cdot h_{sp}) + H_{tl}. \tag{4}$$

$$\forall s \in \mathcal{S}_t : \quad \sum_{p \in \mathcal{P}_s} X_{sp} = 1. \tag{5}$$

$$\forall s \in \mathcal{S}_t, \forall p \in \mathcal{P}_s : \quad X_{sp} \leq h_{sp}. \tag{6}$$

$$\forall s \in \mathcal{S}_t : \quad \sum_{p \in \mathcal{P}_s} h_{sp} \leq K. \tag{7}$$

**Table 2.** Symbols used in the models.

**Parameters**

| | |
|---|---|
| $RG$: | Number of sub-set of requests, $RG \in \{1, 2, \ldots, |\mathcal{S}|\}$. |
| $t$: | Iterator of the global optimization process, $t \in \{1, 2, \ldots, RG\}$. |
| $\mathcal{V}$: | Set of network nodes, $\mathcal{V} = \{v_i : i \in \{1, 2, \ldots, |\mathcal{V}|\}\}$. |
| $\mathcal{L}$: | Set of network links, $\mathcal{L} = \{l_i : i \in \{1, 2, \ldots, |\mathcal{L}|\}$ where $l = (v_i, v_j)$. |
| $\mathcal{J}_t$: | Status of slots at iteration $t$, $\mathcal{J}_t = \{J_{tli} : l \in \mathcal{L}, i \in \{1, \ldots, Fmax\}\}$, if the $i$-th slot of the link $l$ is busy, then $J_{tli} = 1$, otherwise 0. |
| $Fmax$: | Number of slots available per link in the network. |
| $\mathcal{S}$: | Set of requests, $\mathcal{S} = \{s_i : i \in \{1, 2, \ldots, |\mathcal{S}|\}$ where $s = (v_o^s, v_d^s, \lambda^s)$, $v_o^s \in \mathcal{V}$ is source node, $v_d^s \in \mathcal{V}$ is destination node, and $\lambda^s$ is requested bandwidth. |
| $\mathcal{S}_t$: | Sub-set of requests to be installed at iteration $t$, $\mathcal{S}_t \subset \mathcal{S}$. |
| $\Omega_t$: | Sub-set of requests already installed up to iteration $t - 1$, $\Omega_t = \mathcal{S}_1 \cup \ldots \cup \mathcal{S}_{t-1}$. |
| $\mathcal{M}$: | Set of available modulation format, $\mathcal{M} = \{m_i : i \in \{1, 2, \ldots, |\mathcal{M}|\}\}$ where $m = (T, R)$, $T$ is the transmission range and $R$ is the modulation rate. |
| $Kmax$: | Number of available paths per requests. |
| $K$: | Maximum number of traffic sub-flows allowed per request, $1 \leq K \leq Kmax$. |
| $\mathcal{P}_s$: | Set of $Kmax$ paths available for the request $s$, $\mathcal{P}_s = \{p_1, p_2, \ldots, p_{Kmax}\}$ where $p = \{l_1, l_2, \ldots, l_n\}$. |
| $\mathcal{P}_s^l$: | Set of available paths of the request $s$ that use the link $l \in \mathcal{L}$, $\mathcal{P}_s^l \subset \mathcal{P}_s$. |
| $GB$: | Number of slots for guard-band. |
| $C$: | Bandwidth of one slot with BPSK base modulation. |
| $L_l$: | Length of a link $l$, $L_l \geq 0$. |
| $LP_{sp}$: | Length of a path $p \in \mathcal{P}_s$, $LP_{sp} = \sum_{l \in \mathcal{P}_s} L_l$. |
| $\Lambda_{sp}$: | Number of slots assigned to path $p \in \mathcal{P}_s$, $\Lambda_{sp} = \left\lceil \frac{\lambda^s}{R \cdot C} \right\rceil$. |
| $\Lambda_{sm}$: | Number of slots assigned to request $s \in \mathcal{S}$ with modulation $m \in \mathcal{M}$, $\Lambda_{sm} = \left\lceil \frac{\lambda^s}{R_m \cdot C} \right\rceil$. |
| $H_{tl}$: | Number of slots busy in the link $l$ at the iteration $t$, $H_{tl} = \sum_{i=1}^{Fmax} J_{tli}$. |
| $\mu$: | Weight of the sum in objective function in RML phase, $\mu \geq 0$. |
| $\mathcal{L}_v^{out}$: | Set of outgoing links from node $v$, $\mathcal{L}_v^{out} \subset \mathcal{L}$. |
| $\mathcal{L}_v^{in}$: | Set of incoming links to node $v$, $\mathcal{L}_v^{in} \subset \mathcal{L}$. |
| $\Phi_{sp}$: | Number of slots assigned to the already installed request $s \in \Omega_t$ in the $p \in \mathcal{P}_s$, $\Phi_{sp} \in \{0, 1, \ldots, Fmax\}$. |

**Variables**

| | |
|---|---|
| $C_{RML}$: | Maximum slot obtained in the RML phase, $C_{RML} \in \{0, 1, \ldots, Fmax\}$. |
| $C_{SA}$: | Maximum slot obtained in SA phase, $C_{SA} \in \{0, 1, \ldots, Fmax\}$. |
| $B$: | Total number of slots used in the network, $B \in \{0, 1, \ldots, |\mathcal{L}| \cdot Fmax\}$. |
| $F_l$: | Number of slots used in the link $l$, $F_l \in \{0, 1, \ldots, Fmax\}$ |
| $X_{sp}$: | Traffic flow rate of request $s$ on the path $p \in \mathcal{P}_s$, $X_{sp} \in [0, 1]$. |
| $X_{skm}$: | Traffic flow rate of request $s$ on the path $k \in \{1, 2, \ldots, K\}$ with modulation format $m$, $X_{skr} \in [0, 1]$. |
| $h_{sp}$: | Binary variable, if the path $p \in \mathcal{P}_s$ is used, then $h_{sp} = 1$, otherwise 0. |
| $h_{skm}$: | Binary variable, if the path $k \in \{1, 2, \ldots, K\}$ with modulation format $m$ is used, then $h_{skm} = 1$, otherwise 0. |
| $Y_{skml}$: | Binary variable, if the path $k \in \{1, 2, \ldots, K\}$ with modulation format $m$ uses the link $l$, then $Y_{skml} = 1$, otherwise 0. |
| $Z_{skm}$: | Binary variable, if modulation format $m$ is assigned to the path $k \in \{1, 2, \ldots, K\}$, then $Z_{skm} = 1$, otherwise 0. |
| $N_{skm}$: | Number of slots used by the path $k \in \{1, 2, \ldots, K\}$ with modulation format $m$, $N_{skm} \in \{0, 1, \ldots, Fmax\}$. |
| $N_{skml}$: | Number of slots used by the link $l$ of the path $k \in \{1, 2, \ldots, K\}$ with modulation format $m$, $N_{skml} \in \{0, 1, \ldots, Fmax\}$. |
| $f_{sp}$: | First index of a slot block assigned to the path $p \in \mathcal{P}_s$, $f_{sp} \in \{1, 2, \ldots, Fmax\}$. |
| $\delta_{sp}^{s'p'}$: | Binary variable, if $f_{s'p'} > f_{sp}$, then $\delta_{sp}^{s'p'} = 1$, otherwise 0. |

The objective function (1) represents the minimization of the maximum slot used in the network plus the normalized number of used slots.

Equation (2) represents the total number of slots used in the network, while the inequality (3) is a constraint that ensures the maximum slots on each link is less than or equal to $C_{RML}$.

Equation (4) is the maximum slot on each link as consequence of incoming requests plus the maximum slots $H_{tl}$ used on each link by previous installed requests at the iteration $t - 1$. Note that $H_{tl}$ is calculated from the actual network status $\Omega_t$. In the first iteration ($t = 1$) of Algorithm 1, $H_{tl}$ is equal to 0 since the network is empty.

Equation (5) ensures that the total requested traffic flow is allocated to the routes, so that the demand of each request is satisfied. The inequalities (6) and (7) ensure that the number of paths used does not exceed the maximum number of traffic sub-flows.

### 3.4. RML Phase Formulation for MLM

The symbols used in this phase are shown in Table 2. The MLM approach has only one step that differs from the MPM model. The aim is to calculate an RML solution $(Y, N)$ at the iteration $t$ that minimizes $C_{RML}$ the maximum slot, subject to the maximum number of sub-flows $K$, given a sub-set of requests $\mathcal{S}_t$, the state of network resource usage $\mathcal{G}_t = (\mathcal{V}, \mathcal{L}, \mathcal{J}_t)$ and the sub-sets of already installed requests $\Omega_t$. A solution determines the links of routes $Y$ and the number of slots assigned to the links $N$ of each path according to the optimal modulation format, where $Y = \{Y_{skml} : \forall s \in \mathcal{S}, \forall k \in \{1, 2, \ldots, K\}, \forall m \in \mathcal{M}, \forall l \in \mathcal{L}\}$ and $N = \{N_{skml} : \forall s \in \mathcal{S}, \forall k \in \{1, 2, \ldots, K\}, \forall m \in \mathcal{M}, \forall l \in \mathcal{L}\}$. If two solutions have the same $C_{RML}$, the tie is broken by considering the number $B$ of slots used in the network. The objective function (8) has a weighted sum between $C_{RML}$ and the normalized number of slots given by $\frac{B}{|\mathcal{L}|}$ as in objective function (1).

Unlike the previous model (Section 3.3), in this model, all network links can be part of routes. The following formulation expresses the proposed mathematical programming for the first phase of the MLM model.

$$\text{Minimize } \mu \cdot C_{RML} + (1 - \mu) \cdot \frac{B}{|\mathcal{L}|} \tag{8}$$

subject to:

$$B = \sum_{l \in \mathcal{L}} (F_l - GB) \tag{9}$$

$$\forall l \in \mathcal{L}: \quad C_{RML} \geq F_l - GB \tag{10}$$

$$\forall l \in \mathcal{L}: \quad F_l = \sum_{s \in \mathcal{S}_t} \sum_{k=1}^{K} \sum_{m \in \mathcal{M}} N_{skml} + H_{tl} \tag{11}$$

$$\forall s \in \mathcal{S}_t, \forall vs. \in \mathcal{V}: \quad \sum_{k=1}^{K} \sum_{m \in \mathcal{M}} \sum_{v' \in \mathcal{V}} Y_{skm(v,v')} - \sum_{k=1}^{K} \sum_{m \in \mathcal{M}} \sum_{v' \in \mathcal{V}} Y_{skm(v',v)} \leq \begin{cases} K, & \text{if } vs. = v_o^s \\ -K, & \text{if } vs. = v_d^s \end{cases} \tag{12}$$

$$\forall s \in \mathcal{S}_t, \forall k \in \{1, \ldots, K\}, \forall m \in \mathcal{M}, \forall vs. \in \mathcal{V}: \quad \sum_{v' \in \mathcal{V}} Y_{skm(v,v')} - \sum_{v' \in \mathcal{V}} Y_{skm(v,v')} = 0 \tag{13}$$

$$\forall s \in \mathcal{S}_t, \forall k \in \{1, \ldots, K\}, \forall m \in \mathcal{M}, \forall vs. \in \mathcal{V}: \quad \sum_{l \in \mathcal{L}_v^{out}} Y_{skml} \leq h_{skm} \tag{14}$$

$$\forall s \in \mathcal{S}_t, \forall k \in \{1, \ldots, K\}, \forall m \in \mathcal{M}, \forall vs. \in \mathcal{V}: \quad \sum_{l \in \mathcal{L}_v^{in}} Y_{skml} \leq h_{skm} \tag{15}$$

$$\forall s \in \mathcal{S}_t, \forall k \in \{1, \ldots, K\}, \forall m \in \mathcal{M}, \forall vs. \in \mathcal{V}: \quad \sum_{v' \in \mathcal{V}} N_{skm(v,v')} - \sum_{v' \in \mathcal{V}} N_{skm(v,v')} = \begin{cases} N_{skm}, & \text{if } vs. = v_o^s \\ -N_{skm}, & \text{if } vs. = v_d^s \\ 0, & \text{otherwise} \end{cases} \tag{16}$$

$$\forall s \in \mathcal{S}_t, \forall k \in \{1, \ldots, K\}, \forall m \in \mathcal{M}, \forall l \in \mathcal{L}: \quad N_{skml} \leq Y_{skml} \cdot Fmax \tag{17}$$

$$\forall s \in \mathcal{S}_t, \forall k \in \{1, \ldots, K\}, \forall m \in \mathcal{M}, \forall l \in \mathcal{L}: \quad Y_{skml} \leq N_{skml} \tag{18}$$

$$\forall s \in \mathcal{S}_t, \forall k \in \{1,\ldots,K\}: \qquad \sum_{m \in \mathcal{M}} Z_{skm} \leq 1 \tag{19}$$

$$\forall s \in \mathcal{S}_t: \qquad \sum_{k \in \{1,\ldots,K\}} \sum_{m \in \mathcal{M}} X_{skm} = 1 \tag{20}$$

$$\forall s \in \mathcal{S}_t, \forall k \in \{1,\ldots,K\}, \forall m \in \mathcal{M}: \quad X_{skm} \leq Z_{skm} \tag{21}$$

$$\forall s \in \mathcal{S}_t, \forall k \in \{1,\ldots,K\}, \forall m \in \mathcal{M}: \quad \sum_{l \in \mathcal{L}} Y_{skml} \cdot L_l \leq T_m \cdot h_{skm} \tag{22}$$

$$\forall s \in \mathcal{S}, \forall k \in \{1,\ldots,K\}, \forall m \in \mathcal{M}: \quad N_{skm} = \Lambda_{sm} \cdot X_{skm} + GB \cdot h_{skm} \tag{23}$$

$$\forall s \in \mathcal{S}_t, \forall k \in \{1,\ldots,K\}, \forall m \in \mathcal{M}: \quad X_{skm} \leq h_{skm} \tag{24}$$

$$\forall s \in \mathcal{S}_t: \qquad \sum_{k=1}^{k} \sum_{m \in \mathcal{M}} h_{skm} \leq K \tag{25}$$

The objective function (8) minimizes $C_{RML}$ the maximum slot, plus the number $B$ of slots used in the network divided by the number of links in the network, as in (1). Equation (9) represents the number of slots used in the network, while the inequality (10) is a constraint that ensures the maximum slot used on each link is less than or equal to $C_{RML}$.

The constraints (11)–(15) are associated to the routing calculation. Equation (11) represents the maximum slot used on each link by incoming requests plus the maximum slot used on each link by previous requests $H_{tl}$. At the first iteration $t = 1$, $H_{tl}$ is equal to 0 because there is no previous request $\Omega_{t-1} = \varnothing$. Equation (11) adds as many GB as light-paths use the link $l$. However, the GB of the light-path with the highest slot will not be necessary. Consequently, in Equation (9), an adjustment is made, and one GB is subtracted to obtain the exact value of the maximum slot.

Inequality (12) and Equation (13) indicate the links to be used in the path of each request. They also ensure for each request that the number of paths does not exceed the maximum number of traffic sub-flows, and for each path of each request there is only one flow.

Inequality (14) and (15) prevent cycles in the unused paths of each request. The constraints (16)–(18) assign the slots of a link to each path of request. Equation (16) ensures the flow conservation with respect to the slots assigned to each request on each path.

Inequality (17) and (18) ensure that the assigned slots to the links are used to route each path of each request.

The constraints (19)–(25) assign the modulation format to requests. Inequality (19) ensures that just one modulation format is assigned for each path. Equation (20) ensures that total traffic per request is routed. Inequality (21) indicates the modulation format to use for each path of each request. Inequality (22) ensures that the length of each path that uses modulation format cannot be greater than the optical transmission range. Equation (23) represents the number of slots allocated to each path of each request, while inequalities (24) and (25) ensure the number of used paths does not exceed the maximum number of traffic sub-flows.

### 3.5. SA Phase Formulation

Table 2 shows the symbols used in this phase, where the sub-set of requests $\mathcal{S}_t$ to be installed, the sub-sets of already installed requests $\Omega_t$, the state of the network resources $\mathcal{G}_t = (\mathcal{V}, \mathcal{L}, \mathcal{J}_t)$ and the adjacency matrix $Q$ are received as input data. This phase calculates the SA solution $f$ that minimizes $C_{SA}$. An SA solution $f$ determines the first slot assigned to each path of each request $f = \{f_{sp} : \forall s \in \mathcal{S}_t, \forall p \in \mathcal{P}_s\}$.

The following formulation expresses the proposed mathematical programming for this phase, which is used by MLM as well as MPM models.

$$\text{Minimize } C_{SA} \tag{26}$$

subject to:

$$\forall s \in \mathcal{S}_t, \forall p \in \mathcal{P}_s: \quad C_{SA} \geq f_{sp} + \lceil \Lambda_{sp} \cdot X_{sp} \rceil \tag{27}$$

$$\forall \bar{s} \in \Omega_t, \forall p \in \mathcal{P}_{\bar{s}}: \quad C_{SA} \geq f_{\bar{s}p} + \Phi_{\bar{s}p} \tag{28}$$

where $\lceil \Lambda_{sp} \cdot X_{sp} \rceil$ is a constant calculated in the RML phase.

The constraints (29)–(33) are applied iff $Q_{sp}^{s'p'} = 1$, $\forall s \in \mathcal{S}_t, \forall s' \in \mathcal{S}_t, \forall p \in \mathcal{P}_s, \forall p' \in \mathcal{P}_{s'}$:

$$\delta_{sp}^{s'p'} + \delta_{s'p'}^{sp} = 1 \tag{29}$$

$$f_{s'p'} - f_{sp} < \delta_{sp}^{s'p'} \cdot Fmax \tag{30}$$

$$f_{sp} - f_{s'p'} < \delta_{s'p'}^{sp} \cdot Fmax \tag{31}$$

$$f_{sp} + \lceil \Lambda_{sp} \cdot X_{sp} \rceil + GB - f_{s'p'} \leq (Fmax + GB)(1 - \delta_{sp}^{s'p'}) \tag{32}$$

$$f_{s'p'} + \lceil \Lambda_{s'p'} \cdot X_{s'p'} \rceil + GB - f_{sp} \leq (Fmax + GB)(1 - \delta_{s'p'}^{sp}) \tag{33}$$

The constraints (34)–(38) are applied if $Q_{sp}^{\overline{sp}} = 1$, $\forall s \in \mathcal{S}_t, \forall \bar{s} \in \Omega_t, \forall p \in \mathcal{P}_s, \forall \bar{p} \in \mathcal{P}_{\bar{s}}$; note that these constraints are applied among incoming traffic ($s$) and traffic already installed ($\bar{s}$):

$$\delta_{sp}^{\overline{sp}} + \delta_{\overline{sp}}^{sp} = 1 \tag{34}$$

$$f_{\overline{sp}} - f_{sp} < \delta_{sp}^{\overline{sp}} \cdot Fmax \tag{35}$$

$$f_{sp} - f_{\overline{sp}} < \delta_{\overline{sp}}^{sp} \cdot Fmax \tag{36}$$

$$f_{sp} + \lceil \Lambda_{sp} \cdot X_{sp} \rceil + GB - f_{\overline{sp}} \leq (Fmax + GB)(1 - \delta_{sp}^{\overline{sp}}) \tag{37}$$

$$f_{\overline{sp}} + \Phi_{\overline{sp}} + GB - f_{sp} \leq (Fmax + GB)(1 - \delta_{\overline{sp}}^{sp}) \tag{38}$$

The objective function (26) minimizes the maximum slot used in the network.

Inequality (27) ensures that the maximum slot used by incoming requests $s$ is less than or equal to $C_{SA}$. Inequality (28) ensures that the maximum slot used by the previous requests $\bar{s}$ are less than or equal to $C_{SA}$.

Equation (29), and inequalities (30) and (31) indicate the order of the first slot between two incoming requests when they share at least one link. Inequalities (32) and (33) ensure that the slots assigned to the incoming requests do not overlap with each other.

Similarly to the above restrictions, Equation (34), and inequalities (35) and (36) indicate the initial slot between an incoming request $s$ and an already installed request $\bar{s}$ that share at least one link with non-zero load.

The inequalities (37) and (38) ensure that the slot block assigned to an incoming request $s$ as well as an already installed $\bar{s}$ do not overlap each other.

Note that, Equation (28) and inequalities (34)–(38) are deactivated in the first iteration $t = 1$ because the value of $H_{1l} = 0$ as there is no previously installed request and all slots are available to use.

### 3.6. ILP Models Summary

By parameterizing the proposed MPM model, the particular 1PM [29], MP1 and 1P1 [6] models are coded when the number of request sub-sets is set up to one (*RG*=1) and the

number of paths for split traffic is set up to one ($K = 1$). Similarly, by parameterizing the MLM model, the particular 1LM, ML1 and 1L1 [8] models are coded.

A summary of the path-oriented routing-based models can be seen in Table 3, while a summary of the link-oriented-based models can be seen in Table 4. The numbers in the table denote references to the equations explained in the previous sub-sections.

**Table 3.** Summary of equations used by path-oriented routing-based models.

| Phases | Models | | | |
|--------|--------|--------|--------|--------|
| | **MPM** | **1PM** | **MP1** | **1P1** |
| *RML* | (1)–(7) | (1)–(7) | (1)–(7) | (1)–(7) |
| *SA* | (26)–(38) | (26), (27), (29)–(33) | (26)–(38) | (26), (27), (29)–(33) |
| | Model Parameters | | | |
| *RG* | > 1 | = 1 | > 1 | = 1 |
| *K* | > 1 | > 1 | = 1 | = 1 |

**Table 4.** Summary of equations used by link-oriented routing-based models.

| Phases | **MLM** | **1LM** | **ML1** | **1L1** |
|--------|---------|---------|---------|---------|
| *RML* | (8)–(24) | (8)–(24) | (8)–(24) | (8)–(24) |
| *SA* | (26)–(38) | (26), (27), (29)–(33) | (26)–(38) | (26), (27), (29)–(33) |
| | Model Parameters | | | |
| *RG* | > 1 | = 1 | > 1 | = 1 |
| *K* | > 1 | > 1 | = 1 | = 1 |

Note that the RML phase has two possible scenarios while the SA phase has only one.

1. RML phase: the following particular cases should be taken into account

   - *One set of requests*: in Equations (4) and (11), we have that $H_{tl} = 0$ when the value of $\Omega_t = \varnothing$, since there was no previous request; and therefore it represents only the maximum slot used by incoming requests.
   - *Multiple traffic sub-flows*: inequality (12) limits to one the number of paths to be used per request considering $K = 1$. This affects Equation (23), causing it to assign all the required slots to a single path for each request.

2. SA phase:

   - *One set of requests*: Equations (26), (27) and (29)–(33) of the SA phase are used. The other equations are not activated, because $H_{tl} = 0$ and there is no previous request.

To evaluate the complexity of the proposed models, we considered the number of integer variables and restrictions of the proposed models based on the work of [24]. The proportional number of variables, constraints, and asymptotic analysis are presented in Table 5. To obtain the asymptotic of the algorithms, we have considered the following assumptions: the number of modulation formats is constant, the number of candidate routes can grow proportionally to the number of links in the network, and the number of nodes grows proportionally to the number of links in the network.

With the above considerations, we can determine that the path-based RML model introduces a $O(|\mathcal{S}| \cdot |\mathcal{L}|)$ dependency on the number of variables and constraints. The link-based RML model has a $O(|\mathcal{S}| \cdot |\mathcal{L}|^2)$ dependency on variables and constraints. On the other hand, the complexity of SA models is $O(|\mathcal{S}|^2 \cdot |\mathcal{L}|^2)$ depending on variables and constraints. As we can see, for a given network, the number of requests impacts the computing time needed to calculate the solutions.

**Table 5.** Complexity of the proposed models.

| Setting | Variables and Constraints | Proportional Amount | Asymptotic Notation |
|---|---|---|---|
| RML-MPM | Integer Variables | $\propto \|\mathcal{L}\| + \|\mathcal{S}\| \cdot Kmax$ | $O(\|\mathcal{S}\| \cdot \|\mathcal{L}\|)$ |
| | Constraints | $\propto 2 \cdot \|\mathcal{L}\| + 2 \cdot \|\mathcal{S}\| + \|\mathcal{S}\| \cdot Kmax$ | $O(\|\mathcal{S}\| \cdot \|\mathcal{L}\|)$ |
| RML-MLM | Integer Variables | $\propto \|\mathcal{L}\| + 2 \cdot \|\mathcal{S}\| \cdot Kmax \cdot \|\mathcal{M}\| \cdot \|\mathcal{L}\| + 3 \cdot \|\mathcal{S}\| \cdot Kmax \cdot \|\mathcal{M}\|$ | $O(\|\mathcal{S}\| \cdot \|\mathcal{L}\|^2)$ |
| | Constraints | $\propto 2 \cdot \|\mathcal{L}\| + 2 \cdot \|\mathcal{S}\| + 2 \cdot \|\mathcal{S}\| \cdot \|\mathcal{V}\| + \|\mathcal{S}\| \cdot Kmax \cdot \|\mathcal{M}\| \cdot (4 \cdot \mathcal{L} + 6 \cdot \mathcal{V})$ | $O(\|\mathcal{S}\| \cdot \|\mathcal{L}\|^2)$ |
| SA | Integer Variables | $\propto 2 \cdot \|\mathcal{S}\| \cdot Kmax + 2 \cdot \|\mathcal{S}\|^2 \cdot (Kmax)^2$ | $O(\|\mathcal{S}\|^2 \cdot \|\mathcal{L}\|^2)$ |
| | Constraints | $\propto 2 \cdot \|\mathcal{S}\| \cdot Kmax + 2 \cdot \|\mathcal{S}\|^2 \cdot (Kmax)^2$ | $O(\|\mathcal{S}\|^2 \cdot \|\mathcal{L}\|^2)$ |

## 4. Simulation Test

This section studies the performance of the RML+SA models through numerical simulations considering different traffic loads, topologies, and number of requests. This section presents various possible scenarios where the most straightforward procedures help verify the quality of the implemented algorithms. This checking can be done with manual analyses for simple scenarios. On the other hand, the most complex scenarios seek to reproduce real situations of traffic demands. The reasons for these simulations are twofold:

- To determine the benefits of dividing the requests into sub-sets and the advantages in term of computational time of path-based routing over link-based routing.
- To study the performance of the proposed models (MLM, MPM, ML1, MP1, and 1LM) compared to the state-of-the-art models (1L1, 1PM, and 1P1).

### 4.1. Computational Environment

The simulations were performed on a computer with an Intel Core i5-6500 processor (3.20 GHz), 8 GB of DDR4 memory and Windows 10 operating system. The ILP models were implemented in IBM ILOG CPLEX Optimization Studio Version 12.6, and automation using Algorithm 1 was implemented in JAVA 13.

### 4.2. Network Topologies

The tests were performed with the Abeline topology, of 12 nodes and 15 bidirectional links, which can be seen in Figure 2; and the Nobel-eu topology, of 28 nodes and 82 bidirectional links, which can be seen in Figure 3. These topologies belong to the online library SNDlib [44].

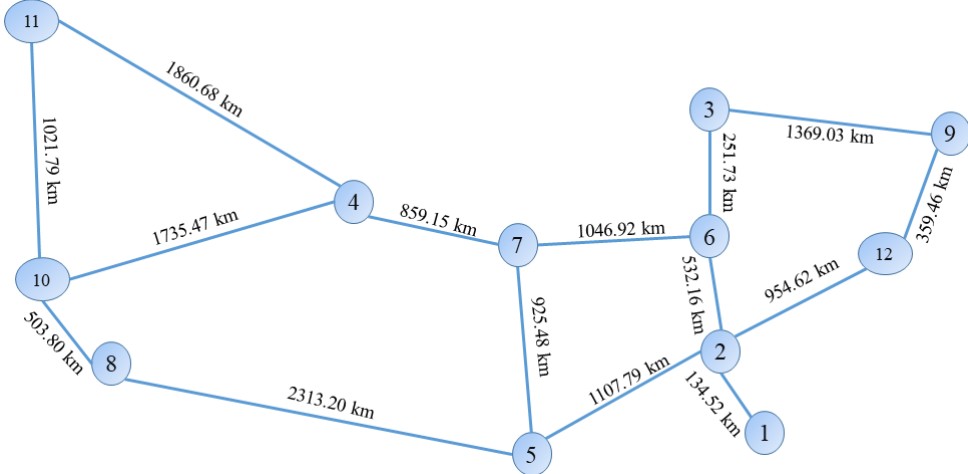

**Figure 2.** Abilene network topology graph.

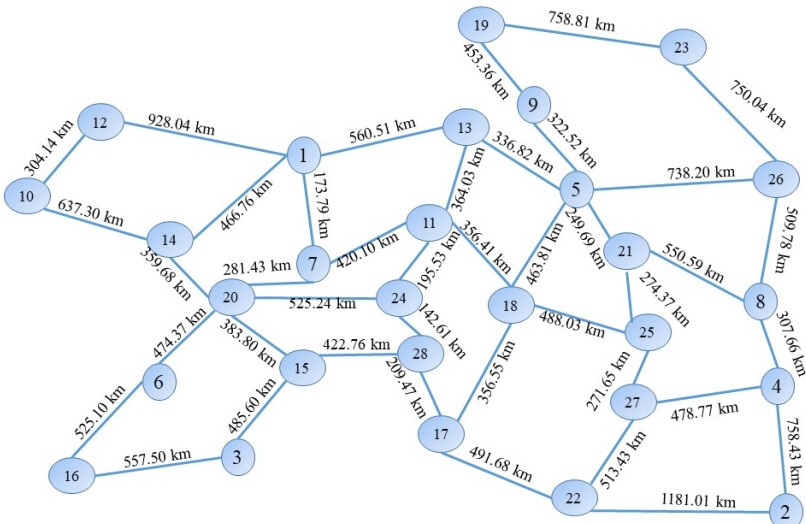

**Figure 3.** Graph of the Nobel-eu network topology.

*4.3. Simulation Scheme*

This sub-section describes the types of simulations performed, as well as the used parameters. For all multiple subsets of request models, the process sorts the requests in descending order according to the requests' cost. The cost of a request is the product of the number of slots and the shortest path length, where the assigned slot block is calculated according to the modulation level.

As general ILP parameters, we consider an optical network with $Fmax$ = 10,000 slots, ward-band $GB$ = 1 slot, and the modulation formats given in Figure 1. Note that we have considered a high value for the slot number to avoid request blocking. The developed scheme considers an invalid solution if any requests are blocked, i.e., the quality of service is outside the scope of this work.

For the Abeline topology, the used traffic loads are the first 40 requests of the Problem abilene–D-B-M-N-C-A-N-N [44], also with a constant load of 625 G bps per request; while for the Nobel-eu topology, the traffic loads used are the first 80 requests of the Problem abilene–D-B-M-N-C-A-N-N, with a constant load of 625 G bps per request. In addition, for the Nobel-eu topology, a computational time limit of 5 minutes was set up for each phase RML and SA, respectively.

Consequently, with these defined parameters and according to the implemented models, the following research questions are raised to conduct the corresponding simulations:

- *Research Question 1*: What would happen as the number of sub-sets of the requests increases as they are divided into smaller sub-sets?
- *Research Question 2*: Are unsplit traffic flow models worse concerning spectrum efficiency and computational time than split traffic flow models? Up to how many sub-demands is convenient to split each request?
- *Research Question 3*: For path-oriented routing, how many available paths ($Kmax$) are advisable to use to improve spectrum-usage efficiency without deteriorating computational time?

  As the available paths ($Kmax$) increase, the spectrum efficiency should increase at the expense of computation time. To answer this question, the performance of solutions were averaged according to the path-oriented routing for the Abeline and Nobel-eu network topologies, considering the following parameters:

  - MP1 ($Kmax = 1, 2, \ldots, 10, K = 1, RG = 5$).
  - MPM ($Kmax = 3, 4, \ldots, 10, K = 3, RG = 5$).
  - 100% load (40 requests for Abeline and 80 requests for Nobel-eu).

- *Research Question 4*: Between the path-oriented and link-oriented routing, which strategy would be the best in terms of used spectrum and computational time?

Link-oriented routing could obtain better results than path-oriented routing in terms of spectrum efficiency at the cost of computational time. To answer this question, the following parameters were considered:

- 1P1 ($Kmax = 4, K = 1, RG = 1$).
- MP1 ($Kmax = 4, K = 1, RG = 3$).
- 1PM ($Kmax = 4, K = 3, RG = 1$).
- MPM ($Kmax = 4, K = 3, RG = 3$).
- 1P1 ($K = 1, RG = 1$).
- ML1 ($K = 1, RG = 3$).
- 1LM ($K = 3, RG = 1$).
- MLM ($K = 3, RG = 3$).
- Loading percentage: $20\%, 40\%, 60\%, 80\%, 100\%$ (8, 16, 24, 32, 40 requests for Abeline and 16, 32, 48, 64, 80 requests for Nobel-eu).

Considering the complexity of the Nobel-eu network topology, the following parameters were also used:

- 1P1 ($Kmax = 8, K = 1, RG = 1$).
- MP1 ($Kmax = 8, K = 1, RG = 3$).
- 1PM ($Kmax = 8, K = 3, RG = 1$).
- MPM ($Kmax = 8, K = 3, RG = 3$).

## 5. Results and Discussion

This section presents the results of the proposed simulation, in order to answer the questions posed in Section 4.3. The corresponding figures associated with these simulations are given in the Appendix A.

### 5.1. Results for Research Question 1

To answer Question 1, the link-oriented and path-oriented routing approaches (MLM and MPM) were tested with the following numbers of request sub-sets $RG = 1, 2, \ldots, 10$. The parameters considered for this simulation include the following: number of available paths $Kmax = 5$, maximum number of traffic sub-flows ($K = 5, 4, 3, 2, 1$), and 100 % traffic load, i.e., $|\mathcal{S}| = 40$ and $|\mathcal{S}| = 80$ requests for Abeline and Nobel-eu topologies, respectively.

Figures A1 and A2 show the box plots of the performance of MLM and MPM versus the number of sub-sets requests, respectively. For each $RG$ value, we can see several solutions corresponding to different $K$. Figures A1a,c and A2a,c show the maximum slot while Figures A1b,d and A2b,d show the computational time.

From this simulation, the results indicate a trade-off between the maximum slot and computational time. In general, when the number of sub-sets $RG$ increases (a) the maximum slot ($C_{sa}$) worsens, while (b) the computational time improves. The first fact is that each group is solved without considering its impact on the successive groups. The second aspect occurs because increasing the number of groups decreases the number of requests and the number of variables in the problem. To measure the trade-off between the maximum slot and the computational time, we have calculated the Pearson correlation, which yields the following values: $-0.289, -0.337, -0.049,$ and $-0.711$ for Abeline with MLM, Abeline with MPM, Nobel-eu with MLM and Nobel-eu with MPM, respectively. Negative values indicate that decreasing the maximum slot implies a longer computation time, which is higher for MLM at Nobel-eu.

In the Abeline topology, the maximum slot and the computational time do not change significantly after $RG > 4$. The computational time of MPM is smaller than MLM due to the fact that MPM explores the whole set of paths. In the Nobel-eu topology, increasing the number of sub-sets does not significantly affect the maximum slot when MPM model is used because it employs the shortest path in all cases and there are sufficient spectrum resources. Regarding the MLM model, the RG affects the maximum slot as the search space increases, and the model fails to converge to an optimal solution. However, the

computational time significantly improves when the set of requests is split with MPM model. Considering $RG > 2$, it is observed that the computational time remains stable. Note that the performance with $RG = 1$ corresponds to 1LM and 1PM approaches, which are particular cases of MLM and MPM, respectively.

*5.2. Results for Research Question 2*

The literature proposes splitting traffic flow to minimize blockages and improve spectrum usage due to optical layer constraints [13,20,29]. However, these reported works do not specify the optimal number of split sub-flows and how it affects the quality of the solutions compared to unsplit traffic in the RML+SA problem.

This simulation aims to describe the behavior of the maximum slot and computational time when the number of traffic sub-flows $K$ varies. The parameters considered for the simulations are the following: number of available paths $Kmax = 5$ and 100% traffic load, i.e., $|\mathcal{S}| = 40$ and $|\mathcal{S}| = 80$ requests for Abeline and Nobel-eu, respectively.

Figures A3 and A4 show the simulation results for Abeline topology while Figures A5 and A6 for Nobel-eu topology. In Figures A3 and A5 was considered only one set $RG = 1$. As $K$ increases, the number of slots improves, and the computational time worsens as the search space grows. For $RG = 1$, we can observe that 1ML and 1PM achieve the same maximum slot while the computational time differs significantly. Figures A3 and A5 show solutions for RG ranging from two to ten subsets of requests. In these figures, we observe a high dispersion of the spectrum used in the solutions for MLM, while it is more stable for MPM. However, for $K > 1$, the spectrum is more convenient. In particular, with $K = 2$, efficient use of the spectrum and less computational time are achieved.

*5.3. Results for Research Question 3*

Figures A7 and A8 show the simulation results, indicating the maximum slot $C_{SA}$ obtained by the path-oriented routing models (MP1 and MPM), considering a number of groups $RG = 5$, and presenting the number of available paths ($Kmax$) in the horizontal axis.

For MP1, we can observe that the maximum slot decreases when the number of available paths $Kmax$ increases. Particularly for Abeline, in Figure A7a, there is a reduction from 380 slots to 305 slots when $Kmax$ goes from 1 to 2. For $Kmax > 2$, the maximum slot remains unchanged since the MP1 model always tends to use the paths that share the least number of links which no longer changes as the number $Kmax$ increases. This results from the fact that in topologies with a low number of nodes and links, the number of disjoint paths for each pair of nodes is reduced; consequently, traffic load balancing is difficult. For the case of Nobel-eu, the increased number of available paths $Kmax$ mainly impacts reducing the maximum slot, see Figure A7b. Nobel-eu is much larger than Abeline; therefore, Nobel-eu presents a higher number of disjoint routes, which helps to balance the traffic load better and reduce the impact of optical layer constraints. This result shows that for $Kmax >= 8$, the maximum slot is also no longer improved since the traffic load balancing is already limited. As for the computational times versus the number $Kmax$, we can observe them in Figure A8a,b. The computational times increase as the number of candidate routes increases for both topologies, i.e., the complexity of the problem increases with $Kmax$. The best performance is observed for $Kmax = 2$ and K = 8 for Abeline and Nobel-eu, respectively.

Let us consider MPM setup with divisible traffic flow up to $Kmax = 3$ sub-flows. Figure A7c,d present the results for the maximum slot, while the computation times are shown in Figure A8c,d, respectively. In Abeline, the MPM scheme has a similar behavior to MP1. Increasing the number of available paths $Kmax$ in a small topology does not impact the improvement of the maximum slot. On the other hand, for the Nobel-eu topology, a similar behavior of the MPM model to MP1 is also observed. The increase of $Kmax$ positively impacts the maximum slot. However, this improvement stagnated after increasing beyond eight routes, $Kmax >= 8$.

*5.4. Results for Research Question 4*

Figures A9 and A10 show box plots of the simulation results for maximum slot and computational time, respectively. This figure shows an overview of all the results considering the configurations analyzed and segmented into link-oriented versus path-oriented approaches.

In general, as the load increases, the needed spectrum grows for both approaches and topologies. This increase in the used spectrum is almost uniform and similar between the path-oriented and link-oriented models, as can be seen in Figure A9. Particularly, for very high load (100% load in the Nobel-eu topology, Figure A9b), a significant difference is observed in the used spectrum considering the path-oriented model with respect to the link-oriented model, due to the fact that it is less efficient in the used spectrum. The computational time used by the link-oriented model is much higher than that used by path-oriented, with this difference being more noticeable for the Nobel-eu topology. From these results, it can be concluded that the use of path-oriented models seems more convenient than link-oriented models when the complexity of the problem increases.

*5.5. General Discussion*

In answering the different research questions, the results obtained guide us to the following conclusions:

- Dividing the requests into multiple subsets up to a certain threshold reduces computation time. The threshold will vary according to the topology considered. Particularly, in both studied topologies, an RG between 3 and 5 is recommended.
- Splitting the traffic stream helps improve the used spectrum at the cost of increased computational time. We recommend splitting the traffic stream into no more than three sub-streams for the studied topologies and loads.
- The number *Kmax* of pre-computed shortest paths may help to improve the efficient use of spectrum and computation time. We observe that three-shortest paths are suitable in the topologies under study.
- Path-based models are more convenient compared to link-based models when the complexity of the problem increases. Determining the appropriate number of routes is necessary to perform simulations since it depends on optical resources and network structure.

From a general perspective, we observe that the MPM model proposed for RML+SA is the most efficient upon considering our study in its entirety.

## 6. Conclusions and Future Work

In this work, we have addressed the routing, modulation level, and spectrum assignment problem under a serialized approach, i.e., RML+SA. Several ILP models were developed by adding request set splitting schemes, and traffic flows with path-oriented and link-oriented routing. The numerical simulations demonstrated the suitability of these approaches, with the path-based method being the more advisable approach. The suitable number of requests and route subsets depends on each topology type and traffic load. Few paths will result in poor spectrum usage, while a large number tends to be computationally expensive without significantly improving spectrum usage. An appropriate number of load set splits helps improve computation time without negatively impacting spectrum usage.

For future work, the authors propose to:

- use the multi-subset request scheme with different heuristics;
- extend this study to the Routing, Baud rate, Code, Modulation Level, and Spectrum Allocation (RBCMLSA) approach;
- use the multi-subset request scheme considering dynamic and reserved traffic; and
- consider other quality metrics such as requests blockage, power consumption, and optical channel impairments.

**Author Contributions:** Conceptualization, D.P.P.-R. and B.B.; methodology, D.P.P.-R.; software, L.V.M.B. and M.M.R.V.P.; validation, J.C. and C.F.B.-M.; formal analysis, B.B.; investigation, C.F.B.-M. and L.V.M.B. and M.M.R.V.P.; data curation, C.F.B.-M., L.V.M.B. and M.M.R.V.P.; writing—original draft preparation, J.C., L.V.M.B. and M.M.R.V.P.; writing—review and editing, D.P.P.-R. and B.B.; visualization, L.V.M.B. and M.M.R.V.P.; All authors have read and agreed to the published version of the manuscript.

**Funding:** This research received no external funding.

**Conflicts of Interest:** The authors declare no conflict of interest.

## Appendix A

This appendix presents the simulation results of Section 5 in Figures A1–A10.

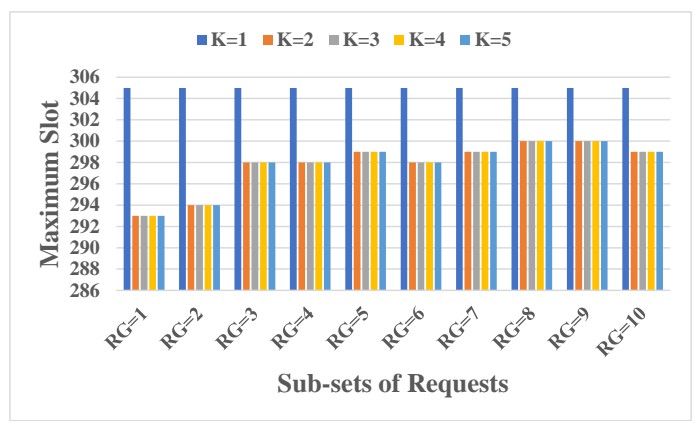

(**a**)

(**b**)

**Figure A1.** *Cont.*

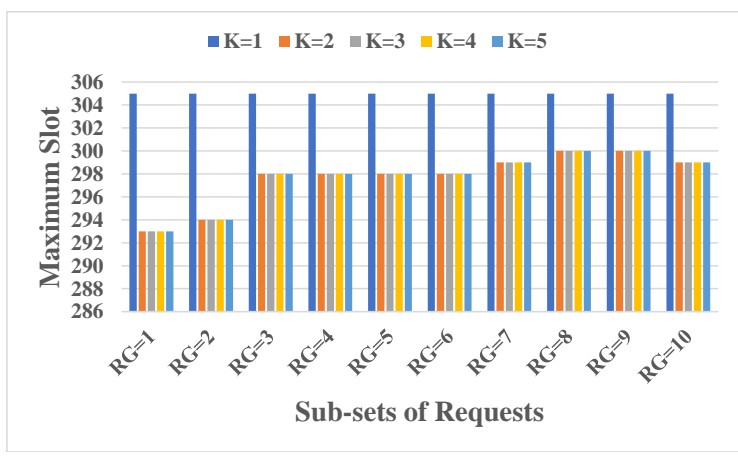

(**c**)

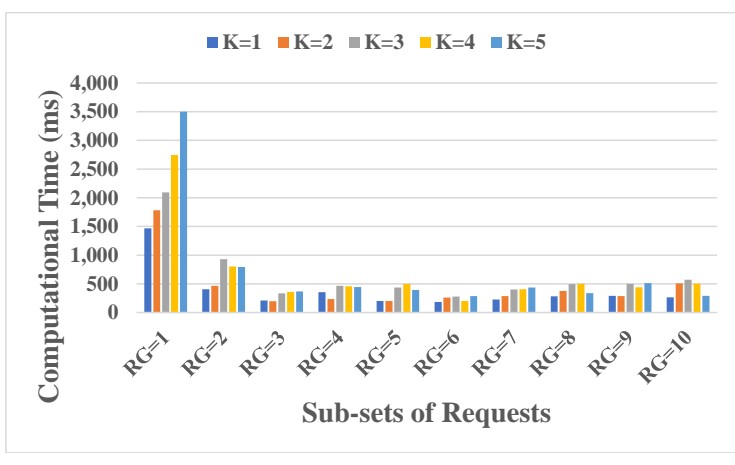

(**d**)

**Figure A1.** Results for *Question 1* Abeline topology. (**a**) Maximum slot in Abeline (MLM); (**b**) Computational time in Abeline (MLM); (**c**) Maximum slot in Abeline (MPM); (**d**) Computational time in Abeline (MPM).

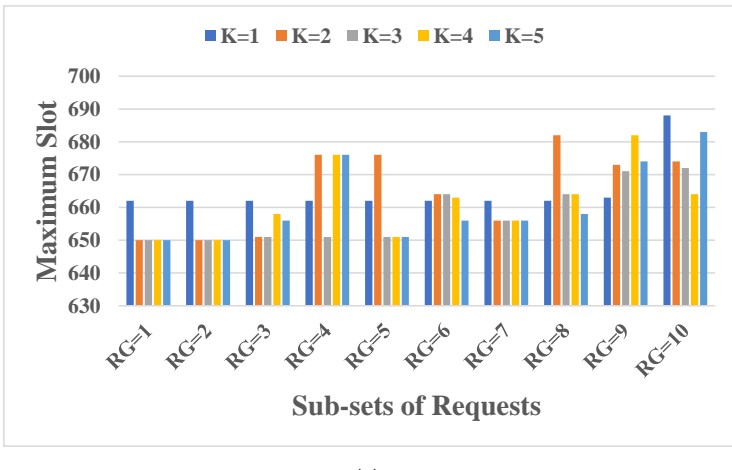

(**a**)

**Figure A2.** *Cont.*

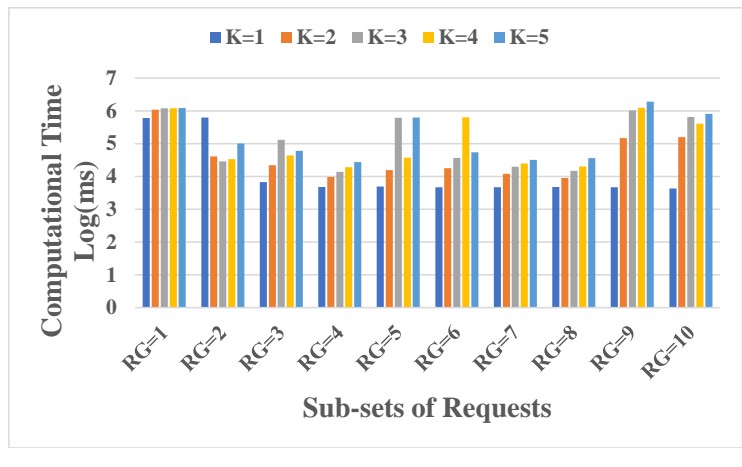

(**b**)

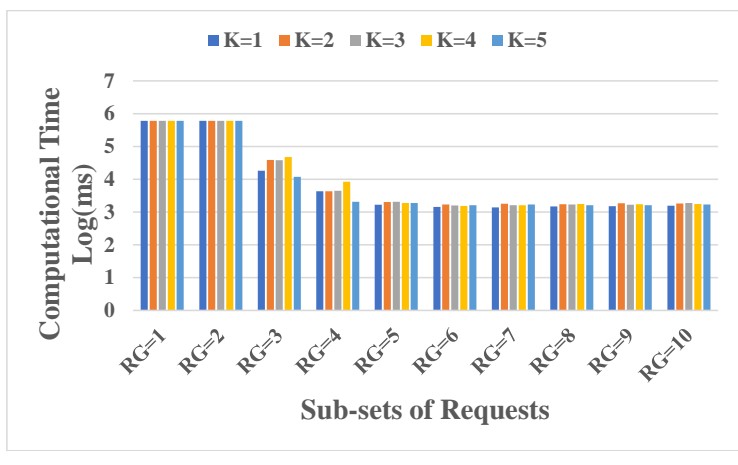

(**c**)

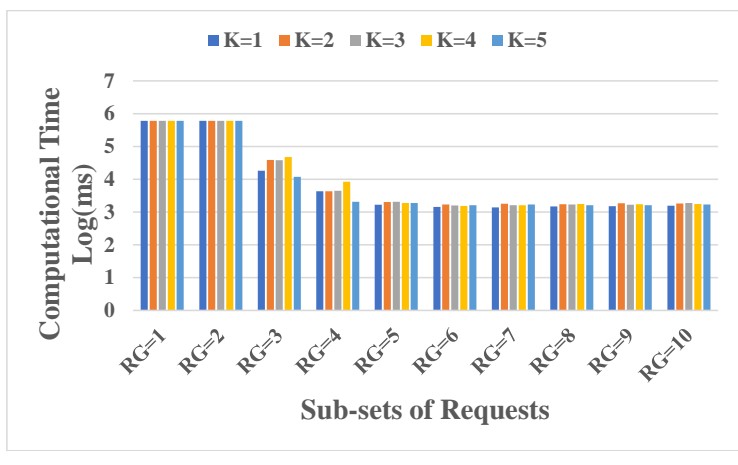

(**d**)

**Figure A2.** Results for Question 1 Nobel-eu topology. (**a**) Maximum slot in Nobel-eu (MLM); (**b**) Computational time in Nobel-eu (MLM); (**c**) Maximum slot in Nobel-eu (MPM); (**d**) Computational time in Nobel-eu (MPM).

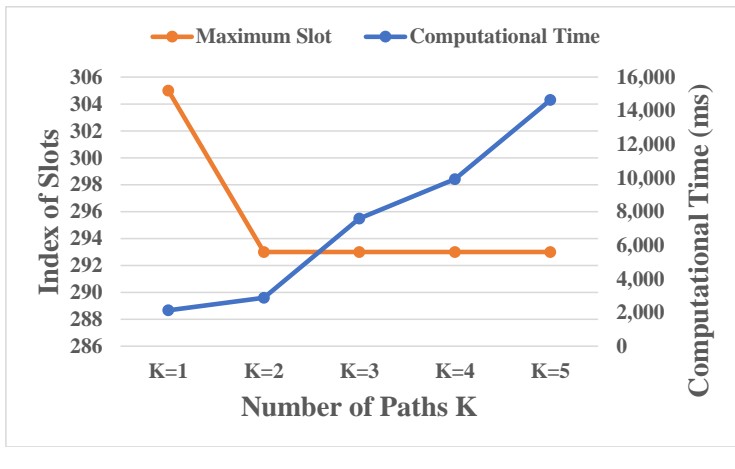

(**a**)

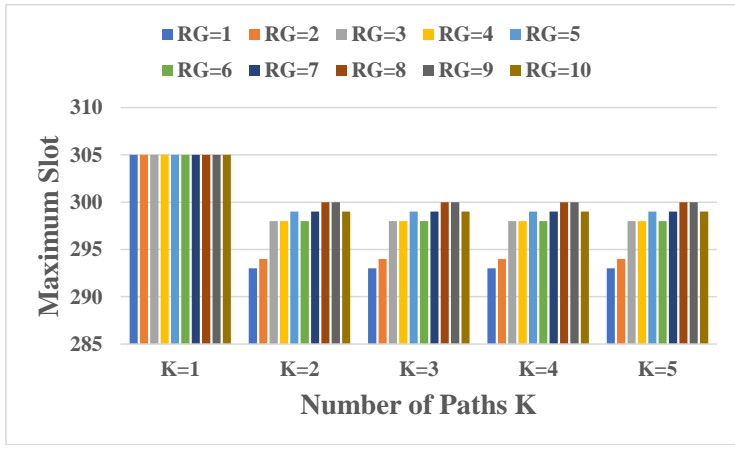

(**b**)

**Figure A3.** Results for Question 2 Abeline topology. (**a**) Maximum slot and Computational Time—1LM; (**b**) Maximum slot and Computational Time—1PM.

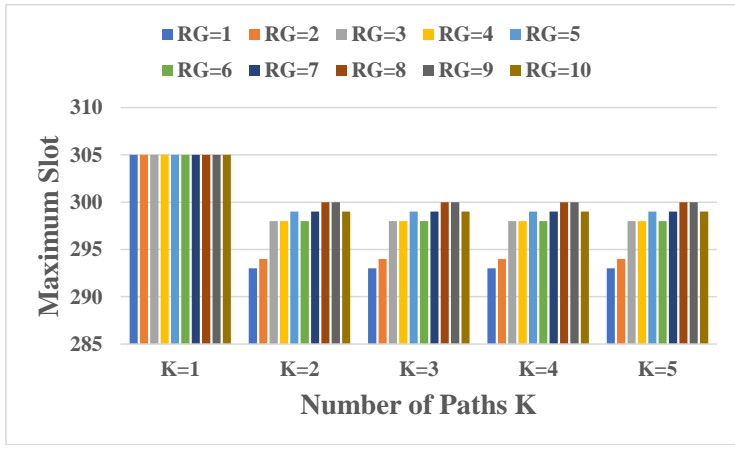

(**a**)

**Figure A4.** *Cont.*

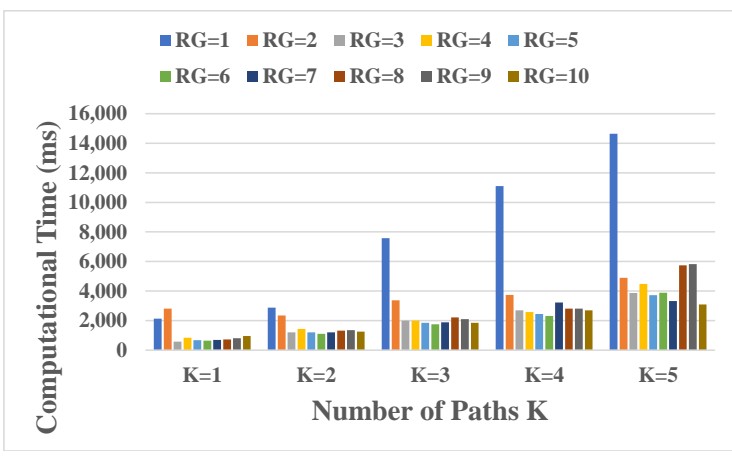

(**b**)

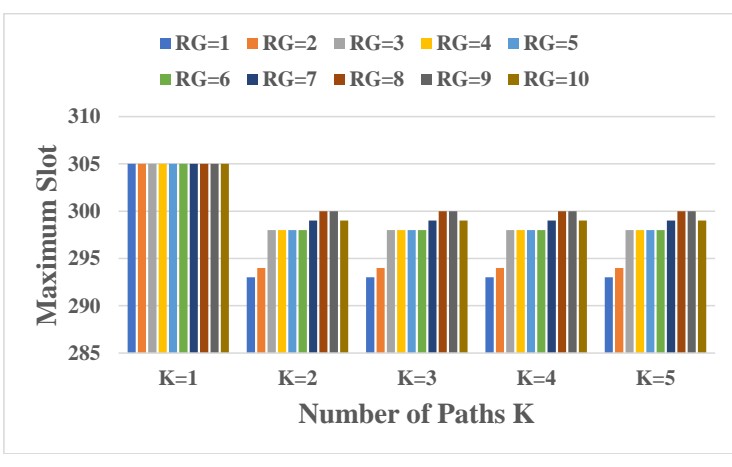

(**c**)

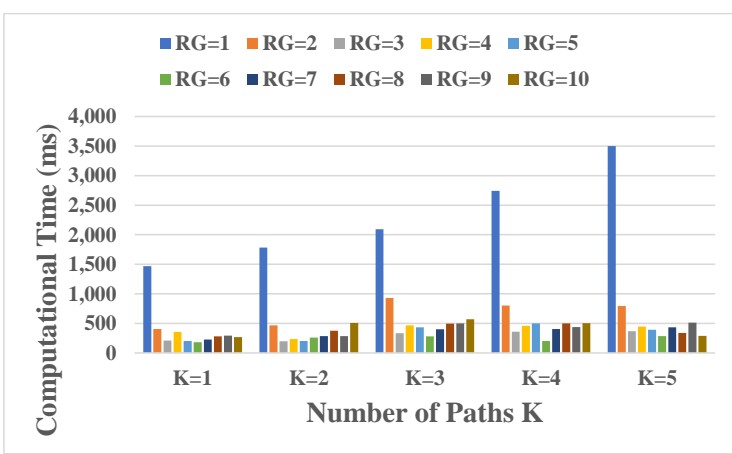

(**d**)

**Figure A4.** Results for Question 2 Abeline topology. (**a**) Maximum slot—MLM; (**b**) Computational time—MLM; (**c**) Maximum slot—MPM; (**d**) Computational time—MPM.

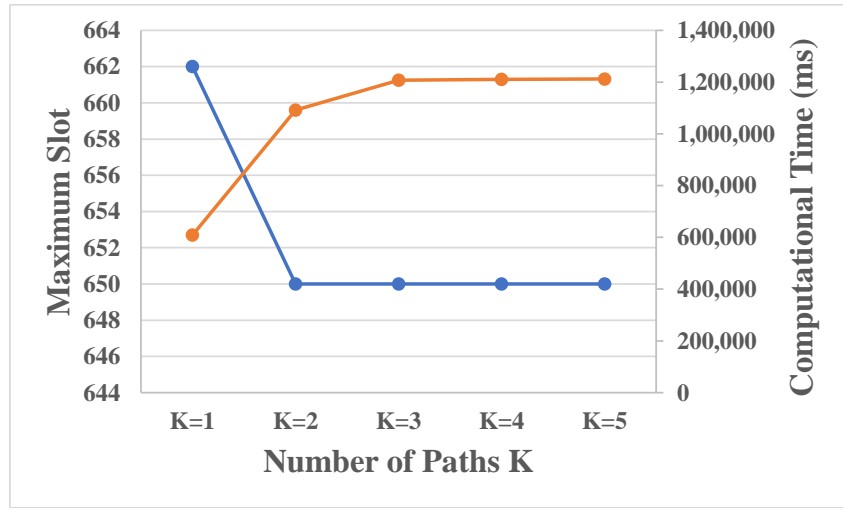

(**a**)

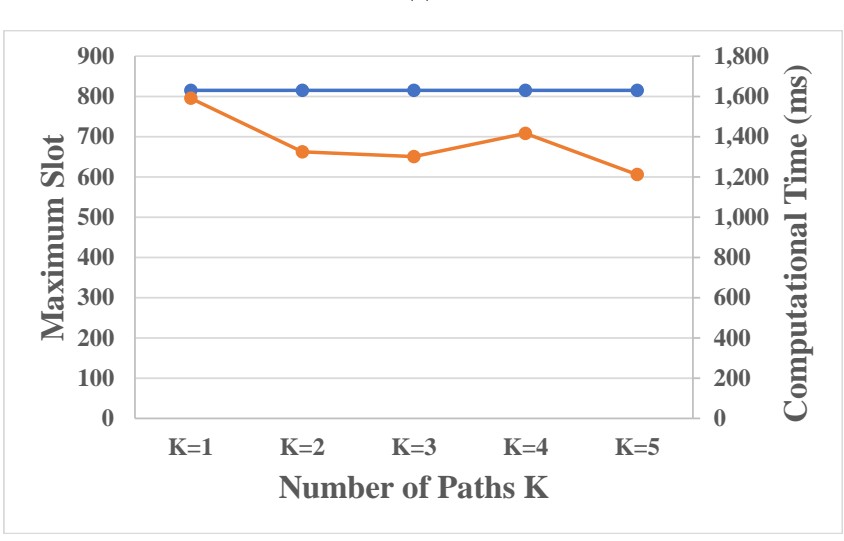

(**b**)

**Figure A5.** Results for Question 2 Nobel-eu topology. (**a**) Maximum slot and Computational Time—1LM; (**b**) Maximum slot and Computational time—1PM.

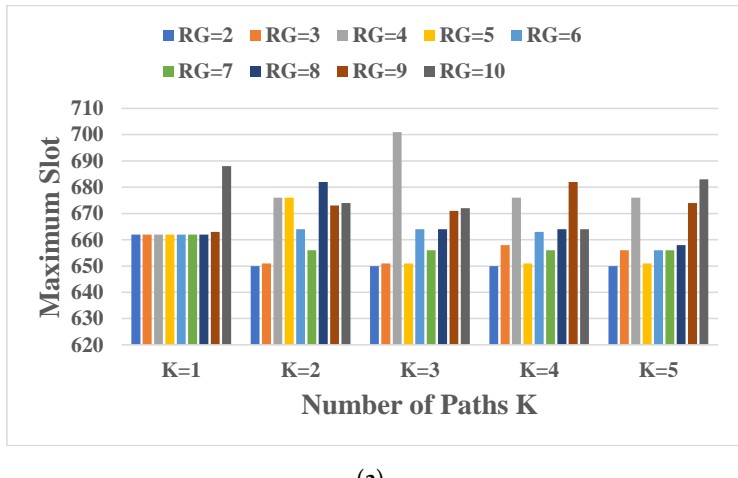

(**a**)

**Figure A6.** *Cont.*

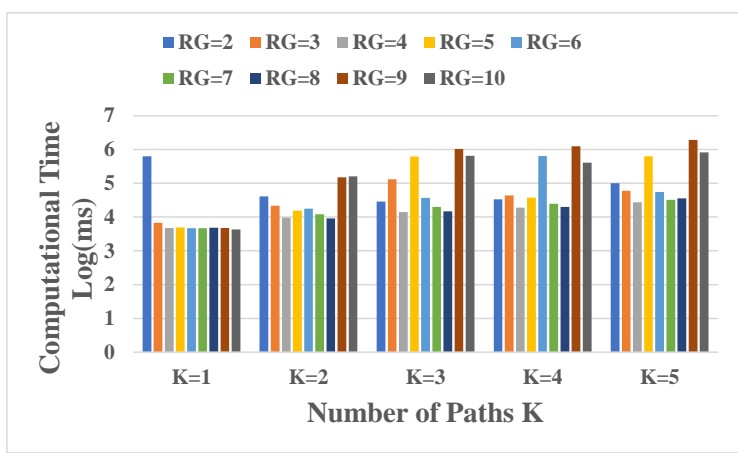

(**b**)

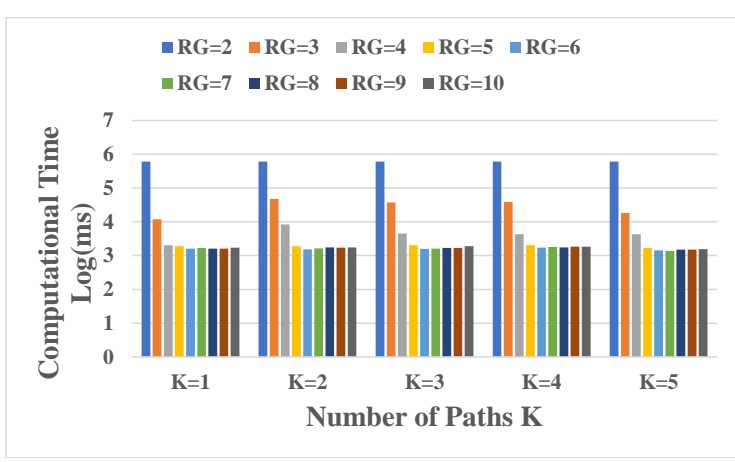

(**c**)

(**d**)

**Figure A6.** Results for Question 2: Nobel-eu topology. (**a**) Maximum slot—MLM; (**b**) Computational time—MLM; (**c**) Maximum slot—MPM; (**d**) Computational time—MPM.

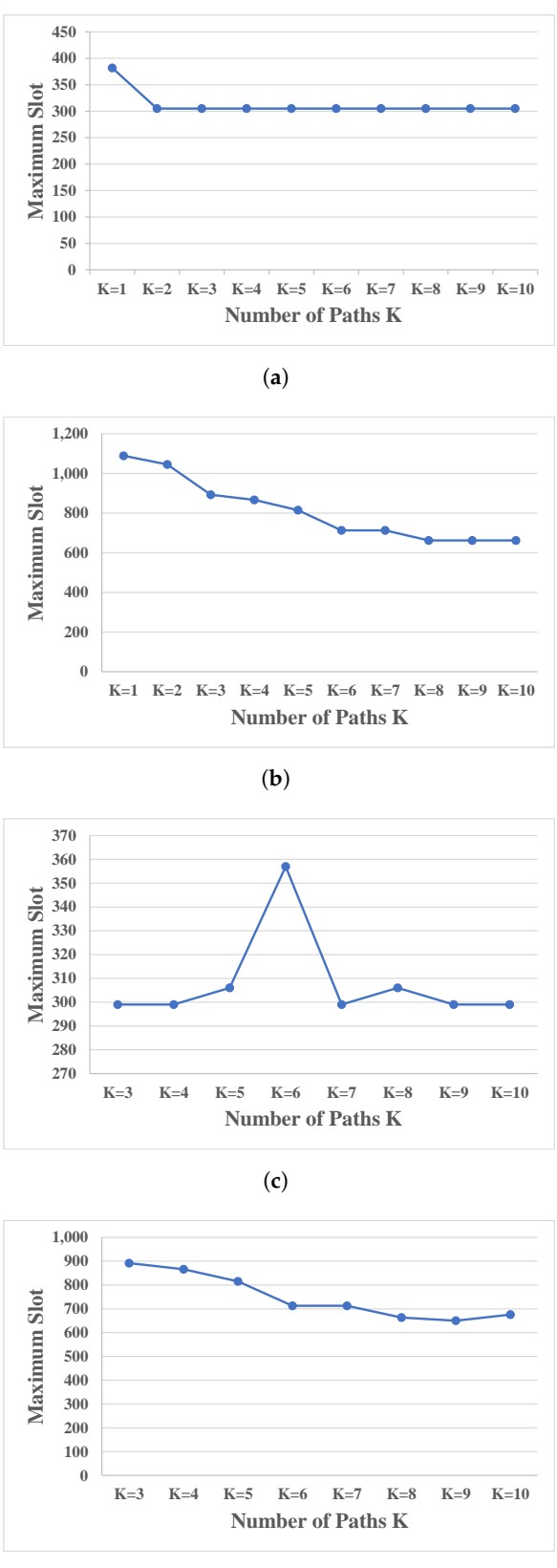

**Figure A7.** Results for Question 3: Abeline and Nobel-eu topologies considering the maximum slot. (**a**) Maximum slot in Abeline (MP1); (**b**) Maximum slot in Nobel-eu (MP1); (**c**) Maximum slot in Abeline (MPM); (**d**) Maximum slot in Nobel-eu (MPM).

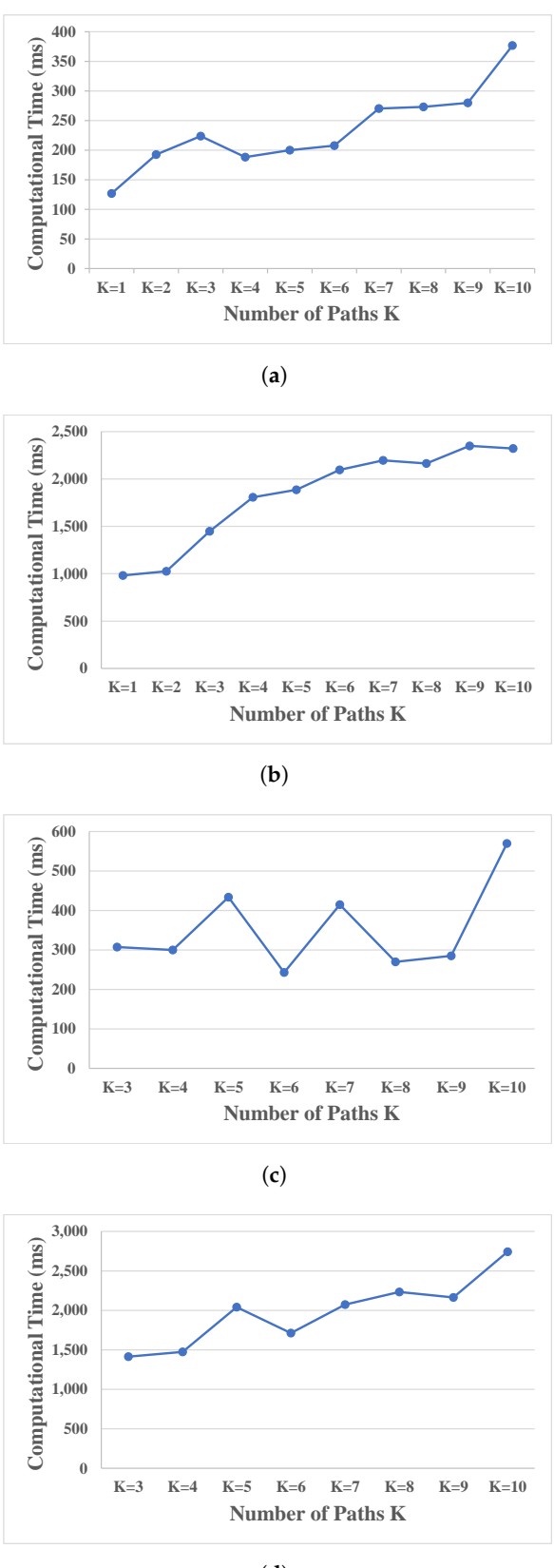

**Figure A8.** Results for Question 3: Abeline and Nobel-eu topologies considering computational time. (**a**) Computational time in Abeline (MP1); (**b**) Computational time in Nobel-eu (MP1); (**c**) Computational time in Abeline (MPM); (**d**) Computational time in Nobel-eu (MPM).

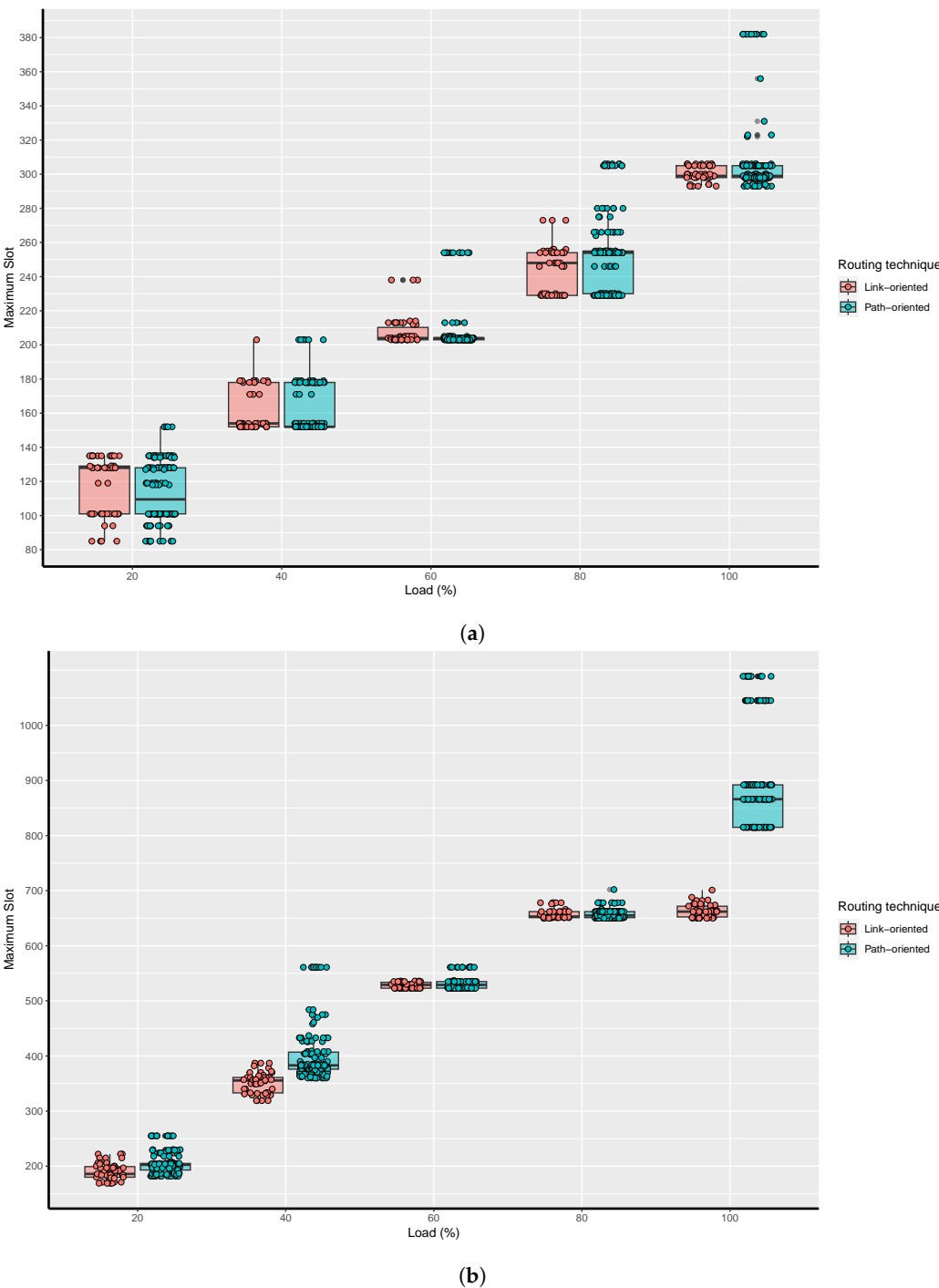

**Figure A9.** Results for Question 4: Maximum slot obtained by path and link oriented models with different traffic load. (**a**) Abeline network topology; (**b**) Nobel-eu network topology.

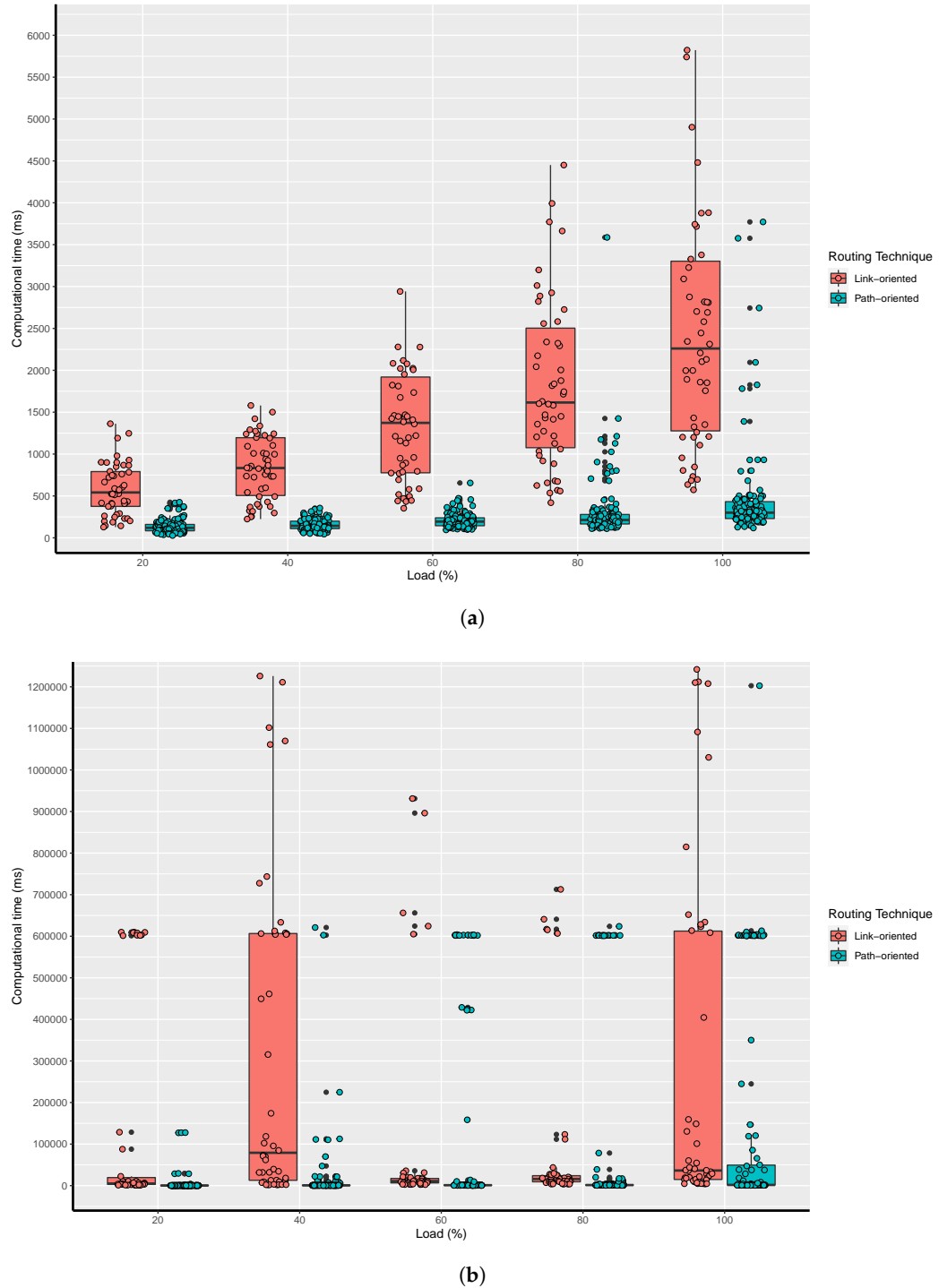

**Figure A10.** Results for Question 4: Computation time obtained by path and link oriented models with different traffic load. (**a**) Abeline network topology; (**b**) Nobel-eu network topology.

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
