# Peer review of "Routing, Modulation Level, and Spectrum Assignment in Elastic Optical Networks—A Serial Stage Approach with Multiple Sub-Sets of Requests Based on Integer Linear Programming"

_mca, doi:10.3390/mca28030067_

Round 1

Reviewer 1 Report

The paper deals with RMLSA problem on elastic optical networks. The authors solved the problem by two phases, the routing and modulation levels are decided first, then spectrums are assigned to each traffic based on the obtained routing and modulation levels. They formulated these problems as integer programming. Especially, they considered multi transmitted paths and scenarios divided a set of traffic demands into several groups. They also performed computational experiments to compare the effect of some parameters and formulation types.

The paper is well written, but the contribution is not so excellent. In particular, the discussion of computing results and conclusions from them lack precision, so it is necessary to present the results that appropriately match the answers to the questions and to provide clear evidence for the conclusions.

Detailed comments:

1.Your review and classification, especially, Table 1 is valuable in this area. However, it is not clear what is the difference between the articles expressed in the same row in Table 1. It is unfortunate that it is not clear what the differences are in the problem setting and solving methods among them.

2. In equations (2) and (3), the description "for all n, m" can be read as a constraint on all node pairs. Here, it should  be "forall mn \in E"

3. When summing for p in equation (3), p must be restricted to pass through the link mn.

4. In the formulation MLM, are variables h and Z equivalent? If these variables are separated, we need a constraint for h corresponding to equation (19).

5. In Table 2, we do not need set of nodes for the path-oriented formulation.

6. In Table 5, the cell for MPM and RML, the equation number is (2) instead of (1).

7. The sentences in the paragraph below in Table 5 seem redundant since they are explained in the previous.

8. The method solving two phases, RML+SA is  heuristic. Although the difference between link-oriented and path oriented formulation was discussed,  it is better to consider the difference from the optimal solutions in  the evaluation of performance.

9. In Figs. A4 and A5, it is better that the results are shown for each parameter K and |SD|. The author say “a trade-off between the maximum FS used and computational time.” However, since we do not know the correspondence between the max FS value and computational time for each instance, this fact can only be assure from the results for different topologies.

10. Why does max FS not monotonically increase as the value of RG increases?

It is wander whether the appropriate value of RG is related to load.

It is not clear How to obtain the conclusion for the optimal AP from Figs. A10 and A11.

Author Response

Response to Reviewer

We thank the suggestions for improvements that were requested. We have tried to introduce best the suggestions indicated by the reviewer.

1.Your review and classification, especially, Table 1 is valuable in this area. However, it is not clear what is the difference between the articles expressed in the same row in Table 1. It is unfortunate that it is not clear what the differences are in the problem setting and solving methods among them.

Answer. Differences between contributions using the same scheme have been introduced in the text, please see lines 131 to 168.

2. In equations (2) and (3), the description "for all n, m" can be read as a constraint on all node pairs. Here, it should  be "forall mn \in E"

Answer. All formulations were adequate according to the suggestion. Please see Table 1 and equations.

3. When summing for p in equation (3), p must be restricted to pass through the link mn.

Answer. In order to present the formulation better, the order of the constraints was slightly modified. Equation (3) is now the equation (4). The suggestion for improvement is correct, since the addition is effectively performed on the paths that use link l. This aspect is considered in the implementation. In table 2 the notation P^{l}_{s} has been added indicating the set of paths that use the link l. This symbol was used in equation (4) and (11) in order to restrict the paths that only use the l link.

4. In the formulation MLM, are variables h and Z equivalent? If these variables are separated, we need a constraint for h corresponding to equation (19).

Answer. Z and h are distinct variables. Equation (19) defines that only one modulation format can be assigned to a request-route pair. While h=1 determines whether there is traffic on a route with assigned modulation. In this sense, equation (25) guarantees that the number of sub-flows does not exceed K.

5. In Table 2, we do not need set of nodes for the path-oriented formulation.

Answer. In the path-oriented formulation, each path is precomputed; in this context, node information is not necessary. In order to clarify this doubt, an explanatory text has been introduced in section 3.3.

6. In Table 5, the cell for MPM and RML, the equation number is (2) instead of (1).

Answer. Table 2, 3 and 4 were merged into table 2 for better readability, since they share much of the same symbology. Table 5 is now table 3. Your suggestion is correct. The index of the equation was adjusted to (1).

7. The sentences in the paragraph below in Table 5 seem redundant since they are explained in the previous.

Answer. Table 5 is now Table 2. This paragraph was deleted and an introduction to the possible scenarios of the RML and SA phases was added.

8. The method solving two phases, RML+SA is  heuristic. Although the difference between link-oriented and path oriented formulation was discussed,  it is better to consider the difference from the optimal solutions in  the evaluation of performance.

Answer. This inquiry was analyzed in Question 4 for different configurations. The simulation results of this question are given in section 5.4.

9. In Figs. A4 and A5, it is better that the results are shown for each parameter K and |SD|. The author say “a trade-off between the maximum FS used and computational time.” However, since we do not know the correspondence between the max FS value and computational time for each instance, this fact can only be assure from the results for different topologies.

Answer. All figures have been modified in order to identify the RG and K in them. In this sense, it will be possible to observe the variations depending on K and RG.

10.a.  Why does max FS not monotonically increase as the value of RG increases?

Answer. The Figures were modified in order to observe this trend, for example Figure A4.a increases as RG increases for values K>1. The reverse is observed for computational time.

10.b. It is wander whether the appropriate value of RG is related to load.

Answer. In these simulations the value of RG was taken randomly. The appropriate value of RG is based on the structure of the network. These claims were included in section 6.

10.c It is not clear How to obtain the conclusion for the optimal AP from Figs. A10 and A11.

Answer. We introduce better explanations in this section by explaining the behavior of the maximum slot utilization and the computation time as a function of the number of available routes. We also explain which is the number of routes with the best performance.

Reviewer 2 Report

This paper proposed RML+SA solutions for elastic optical networks considering multiple sequential sub-sets of requests, split traffic flow, as well as path-oriented and link-oriented routing models. And numerical simulations have shown the suitability of these approaches. However, some concerns should be addressed before its publication.

1. The title of the paper should be revised, due to the grammar mistakes.

2. Given the request splitting, the authors should explain if there is any extra requirement for hardware to realize such splitting, which may possibly reduce the feasibility of the proposed solutions in reality.

3. In the introduction part, the authors should not count “Review and classification of approaches based on ILP techniques to solve the off-line RMLSA problem” as one of the contribution of the paper.

4. Grammar mistake can be found in the line 171 and 172.

5. In the Mathematical Programming Models section of the paper, the authors should give the complexity computation for the algorithm.

6. In the simulation test section of the paper, I wonder how the results change when different number FSs per request, such 40 FSs, are adopted in the simulation.

Author Response

Response to Reviewer

We thank the suggestions for improvements that were requested. We have tried to introduce best the suggestions indicated by the reviewer.

1. The title of the paper should be revised, due to the grammar mistakes.

Answer. The title was adjusted.

2. Given the request splitting, the authors should explain if there is any extra requirement for hardware to realize such splitting, which may possibly reduce the feasibility of the proposed solutions in reality.

Answer. In this work we assume that the computational resources are sufficient and such computations do not affect the optimization process. This assumption has been introduced in section 3.2 since we do not expect a very high number of requests involving a large time in the ordering prior to the division of the set into subsets. On the other hand, we also assume that the technology used to perform the traffic flow division must be used as OFDM Sliceable width transponders. Please see line 204 to 207.

3. In the introduction part, the authors should not count “Review and classification of approaches based on ILP techniques to solve the off-line RMLSA problem” as one of the contribution of the paper.

Answer. We agree with the suggestion, therefore the line was deleted from the contribution.

4. Grammar mistake can be found in the line 171 and 172.

Answer. This was corrected.

5. In the Mathematical Programming Models section of the paper, the authors should give the complexity computation for the algorithm.

Answer.Model complexity analysis was added in section 3.6. Please see lines 384 to 390 as well as Table 5.

6. In the simulation test section of the paper, I wonder how the results change when different number FSs per request, such 40 FSs, are adopted in the simulation.

Answer.This study analyzes the static traffic scenario considering planning aspects. In this sense, the blocking rate has not been incorporated in the mathematical model as an element to be minimized in the objective function. As the number of frequency slots decreases, it is possible that requests will be blocked as optical resources tend to become scarce. We incorporate this inquiry as future work in the conclusions section. This clarification was added in line 427.

Reviewer 3 Report

The paper addresses serialized approaches of the routing, modulation level, and spectrum assignment (RMLSA) problem in elastic optical networks. RMLSA is split into RML and SA phases and several versions of this problem are investigated using Integer Linear Programming (ILP) models. Overall the paper is well written and the methodology used is adequately described. My main concern regarding this paper is the results section: the plots itself, the comments on the results and some conclusions drawn by the authors from the results. The questions raised by the authors are interesting but in my point of view it is very hard to answer such questions using the graphs presented by the authors in the paper. The section 5 requires a major revision.

I expect to observe a solid trend (either increasing or decreasing in y axis value as the x axis increases) in the majority of the results shown by the authors. Instead,in the majority of the graphs one observes either a flat or a random trend on the boxes' means. The author needs to explain that. Maybe the number of experiments used to build each box is not enough to guarantee the "convergence" (I mean the trend convergence not the RMLSA convergence) and formation of such trend.  

"Multiple sub-sets of requests: The set of requests is divided into several smaller sub-sets, ..."  - The concept of "Multiple sub-sets of request" requires a better explanation. Moreover, is this concept linked with some practical aspect faced by carriers? or is it just a form of easing the computational solution for the RMLSA problem? Please answer and contextualize it in the manuscript. 

Please give a better motivation (in section 2) why are you considering a two-stage RML+SA problem. Give its pros and cons, because it clearly performs worse than one stage RMLSA.

"... to cover cases still not considered, as indicated in Table 1 with shaded cells." - Why is it important to cover such cases? Please motivate it in the manuscript.

"Two-stage serial optimization: In this approach, algorithms" - Please mention in the text that there are also works that solves the Two-stage problem in SA+RML order:

https://doi.org/10.1016/j.osn.2013.09.003

https://doi.org/10.1016/j.comnet.2021.108287

"models 1LM, ML1, and MP1; as well as models already reported in the literature for 1L1 [8], 1PM [17], and 1P1 [6]" - Recent papers are missing for either 1PM and 1LM (please include): 

https://doi.org/10.1016/j.comnet.2021.107895

https://doi.org/10.1016/j.yofte.2022.103208

"(a) mono-objective approaches [5,6,8,13,16,17,19–21,23,27,28,31,32] " - More recent references than ones provided in the manuscript for mono-objective RSA: 

https://doi.org/10.1016/j.osn.2018.05.001

https://doi.org/10.1016/j.comnet.2022.109478

https://doi.org/10.3390/math10183293

Please include.

"Each SD_i is an input for the MILP or ILP, …" Does it mean that you are considering that different models may be applied for different values of i? Please clarify in the text.

If the maximum reach for BPSK is 1000 km (as shown in c) why the request v_3 (1300 km) is using BPSK (as shown in d)? v_3 is impossible to be established considering the scenario shown in Fig 1.

Table 2 and Table 3, C_{RML} - Maximum index of FS obtained in RML phase. How can you know/evaluate, during the RML phase, the maximum index of FS if the SA was not solved yet? Please explain the approximation that you are considering to evaluate C_{RML} during RML (in this sense, maybe you are assuming full spectrum conversion at nodes). It would be helpful if the authors provided the mathematical formula used to evaluate C_{RML}.

Eq. 9 - Why are you subtracting GB in this equation? The frequency slots occupied by the guard bands should be also considered as a part of the spectral occupancy.  Please explain the reasons in the paper. 

The actual design variables in table 4 are the two \delta variables, right? If yes, the term "indicator" is confusing, if not, maybe the actual design variables are missing in this table.

"and the advantages of path-based routing over link-based routing" - Please be careful, this is a misleading statement, only in very specific cases this statement might be true. The set of solutions that can be found by a path-based routing model is always a subset of the solution set achieved by a link-based routing model.

"we consider an optical network with 10000 FSs.". Why this number? The usual number of FS are 320 for slots of 12,5GHz wide and 640 for slots of 6,25GHz wide. Please justify in the paper.  

"constant load of 50 FSs per request" - If you are performing the modulation level assignment, the number of frequency slots used by each request is variable depending on the route length. Maybe you mean 50 Gb/s. Please clarify.     

Further explanation on box plots shown in the graphs are required in the manuscript. Boxplot is a method for graphically demonstrating the spread and skewness of groups of numerical data through their quartiles. What is changing in your algorithm so you can find several different data (result) to build each box plot? How many trials are done? Which is the random variable (and its distribution) linked with these trials? Also, please explain how you represent mean, median, quartiles, outliers in each box.    

"For each RG value, we can see several solutions corresponding to different K." - I cannot see it in the graph. Where is K?

"From this simulation, the results indicate a trade-off between the maximum FS used and computational time. … (a) the maximum FS used (Csa) worsen, while (b) the computational time improves" - I cannot observe the mentioned trend for Csa and computational time in the graphs. A careful explanation is required, please explain in te paper the trend (or not)observed in each graph.

"However, they do not specify the optimal number of split sub-flows and how it affects the quality of the solutions compared to un-split traffic."- There are several papers in the literature that report the mentioned analysis.

" As K increases, the number of used FS improves, and the computational time worsens as the search space grows​​" - ​​Again, I cannot observe the mentioned trend in the graphs: FS is almost constant regardless of the K. There are some exceptions in which the mentioned trend is observed. Therefore, these graphs should be analyzed one by one. 

"…res A10 and A11 show the simulation results, indicating the maximum used FS obtained by the path-oriented routing models (MP1 and MPM), considering a number of requests groups RG = 5, and presenting the number of available paths (K) in the horizontal axis." - It is written AP in the figures instead of  K. Please correct.

In my point of view, all conclusions drawn in section "5.5. General Discussion" are incomplete, inadequate or misleading. None of the conclusions can be drawn directly from the graphs shown in the paper. Particularly, the conclusion in the last bullet cannot be given without the proper context.

Author Response

Response to Reviewer

We thank the suggestions for improvements that were requested. We have tried to introduce best the suggestions indicated by the reviewer.

  1. The paper addresses serialized approaches of the routing, modulation level, and spectrum assignment (RMLSA) problem in elastic optical networks. RMLSA is split into RML and SA phases and several versions of this problem are investigated using Integer Linear Programming (ILP) models. Overall the paper is well written and the methodology used is adequately described. My main concern regarding this paper is the results section: the plots itself, the comments on the results and some conclusions drawn by the authors from the results. The questions raised by the authors are interesting but in my point of view it is very hard to answer such questions using the graphs presented by the authors in the paper. The section 5 requires a major revision.

Answer. Section 5 has been improved both in explanations and in the quality of the figures that help to understand the explanations.

  1. I expect to observe a solid trend (either increasing or decreasing in y axis value as the x axis increases) in the majority of the results shown by the authors. Instead,in the majority of the graphs one observes either a flat or a random trend on the boxes' means. The author needs to explain that. Maybe the number of experiments used to build each box is not enough to guarantee the "convergence" (I mean the trend convergence not the RMLSA convergence) and formation of such trend.  

Answer. We agree with the suggestions. The Figures were modified in order to clearly observe the trends.

  1. "Multiple sub-sets of requests: The set of requests is divided into several smaller sub-sets, ..."  - The concept of "Multiple sub-sets of request" requires a better explanation. Moreover, is this concept linked with some practical aspect faced by carriers? or is it just a form of easing the computational solution for the RMLSA problem? Please answer and contextualize it in the manuscript.  Esta aclaracion fue agregada en la seccion 3.1

Answer. This clarification was added in section 3.1.

  1. Please give a better motivation (in section 2) why are you considering a two-stage RML+SA problem. Give its pros and cons, because it clearly performs worse than one stage RMLSA.

Answer. In that section it was added the reasons why RML+SA achieves a solution in less time compared to RMLSA but sacrificing the quality of the solution.

  1. "... to cover cases still not considered, as indicated in Table 1 with shaded cells." - Why is it important to cover such cases? Please motivate it in the manuscript.

Answer. The requested motivation was added.

  1. "Two-stage serial optimization: In this approach, algorithms" - Please mention in the text that there are also works that solves the Two-stage problem in SA+RML order:

https://doi.org/10.1016/j.osn.2013.09.003

https://doi.org/10.1016/j.comnet.2021.108287

Answer. The works have been added and indicated in the paper.

  1. "models 1LM, ML1, and MP1; as well as models already reported in the literature for 1L1 [8], 1PM [17], and 1P1 [6]" - Recent papers are missing for either 1PM and 1LM (please include): 

https://doi.org/10.1016/j.comnet.2021.107895

https://doi.org/10.1016/j.yofte.2022.103208

Answer. The works have been added.

  1. "(a) mono-objective approaches [5,6,8,13,16,17,19–21,23,27,28,31,32] " - More recent references than ones provided in the manuscript for mono-objective RSA: 

https://doi.org/10.1016/j.osn.2018.05.001

https://doi.org/10.1016/j.comnet.2022.109478

https://doi.org/10.3390/math10183293

Please include.

Answer. All suggestions were included in the corresponding sections.

  1. "Each SD_i is an input for the MILP or ILP, …" Does it mean that you are considering that different models may be applied for different values of i? Please clarify in the text.

Answer. This part was corrected… ”is an input for the RML phase”.

  1. If the maximum reach for BPSK is 1000 km (as shown in c) why the request v_3 (1300 km) is using BPSK (as shown in d)? v_3 is impossible to be established considering the scenario shown in Fig 1.

Answer. The graphic example was corrected by adjusting the route distance to the maximum range indicated.

  1. Table 2 and Table 3, C_{RML} - Maximum index of FS obtained in RML phase. How can you know/evaluate, during the RML phase, the maximum index of FS if the SA was not solved yet? Please explain the approximation that you are considering to evaluate C_{RML} during RML (in this sense, maybe you are assuming full spectrum conversion at nodes). It would be helpful if the authors provided the mathematical formula used to evaluate C_{RML}.

Answer. The suggestion for improvement is correct. In the RML phase the maximum FS index is obtained from the RML problem but not from the entire RML+SA problem. The clarification has been added in sections 3.3, 3.4 and in the Symbol Table.

  1. Eq. 9 - Why are you subtracting GB in this equation? The frequency slots occupied by the guard bands should be also considered as a part of the spectral occupancy.  Please explain the reasons in the paper. 

Answer. In equation (11) the total number of FS including one GB per route is taken into account. Consequently, the GB of the path with the highest index should not consider this GB, so an adjustment is made in equation (9). Please see lines 310 to 313.

  1. The actual design variables in table 4 are the two \delta variables, right? If yes, the term "indicator" is confusing, if not, maybe the actual design variables are missing in this table.

Answer. For the spectrum assignment model the design variable is "f" while the delta variable is an auxiliary variable. In the table the indicator term was adjusted by a Binary Variable.

  1. "and the advantages of path-based routing over link-based routing" - Please be careful, this is a misleading statement, only in very specific cases this statement might be true. The set of solutions that can be found by a path-based routing model is always a subset of the solution set achieved by a link-based routing model. 

Answer. The reviewer's statement is entirely correct. We have adjusted the text indicating that the advantages are in terms of computation time. Please see lines 583 to 586.

  1. "we consider an optical network with 10000 FSs.". Why this number? The usual number of FS are 320 for slots of 12,5GHz wide and 640 for slots of 6,25GHz wide. Please justify in the paper.  

Answer. The models considered were developed under the assumption that the resources are sufficient for all requests to be served. In this sense, we have used a high value to guarantee this assumption since we have used traffic reported from SNDlib. This clarification was added in the corresponding paragraph. Please see lines 425 to 428.

  1. "constant load of 50 FSs per request" - If you are performing the modulation level assignment, the number of frequency slots used by each request is variable depending on the route length. Maybe you mean 50 Gb/s. Please clarify. 

Answer. The reviewer's statement is correct. The text was adjusted according to the reviewer's indications, please see line 430.

  1. Further explanation on box plots shown in the graphs are required in the manuscript. Boxplot is a method for graphically demonstrating the spread and skewness of groups of numerical data through their quartiles. What is changing in your algorithm so you can find several different data (result) to build each box plot? How many trials are done? Which is the random variable (and its distribution) linked with these trials? Also, please explain how you represent mean, median, quartiles, outliers in each box.  

Answer. The graphs were modified in order to improve reading comprehension.

  1. "For each RG value, we can see several solutions corresponding to different K." - I cannot see it in the graph. Where is K?

Answer. The graphs were modified to identify the value of K and RG for each solution.

  1. "From this simulation, the results indicate a trade-off between the maximum FS used and computational time. … (a) the maximum FS used (Csa) worsen, while (b) the computational time improves" - I cannot observe the mentioned trend for Csa and computational time in the graphs. A careful explanation is required, please explain in te paper the trend (or not)observed in each graph.

Answer. To help the reader understand this trend, we have added Pearson correlation as a measure of trade-off, which indicates that negative values imply that improvement in one criterion is achieved with the worsening of another criterion. See lines 491 to 495 and Figures A6 and A8, which show these trends.

  1. "However, they do not specify the optimal number of split sub-flows and how it affects the quality of the solutions compared to un-split traffic."- There are several papers in the literature that report the mentioned analysis.

Answer. None of the works reported in Table 1 have reported said study for RML+SA. We include that observation in the text. Please see lines 128 to 130.

  1. " As K increases, the number of used FS improves, and the computational time worsens as the search space grows​​" - ​​Again, I cannot observe the mentioned trend in the graphs: FS is almost constant regardless of the K. There are some exceptions in which the mentioned trend is observed. Therefore, these graphs should be analyzed one by one. 

Answer. The Figures were modified, in particular, Figures A6a, A6b, A8a and A8b were adjusted to identify commitment relationships. In the case of Figure A8b the trade-off relationship is not fulfilled.

  1. "…res A10 and A11 show the simulation results, indicating the maximum used FS obtained by the path-oriented routing models (MP1 and MPM), considering a number of requests groups RG = 5, and presenting the number of available paths (K) in the horizontal axis." - It is written AP in the figures instead of  K. Please correct.

Answer. The symbol has been adjusted. K instead of AP.

  1. In my point of view, all conclusions drawn in section "5.5. General Discussion" are incomplete, inadequate or misleading. None of the conclusions can be drawn directly from the graphs shown in the paper. Particularly, the conclusion in the last bullet cannot be given without the proper context.

Answer. The discussions were adjusted throughout section 5.5 in consideration of the changes requested by the reviewers.

Round 2

Reviewer 1 Report

The manuscript has been improved and it is now good condition.

Reviewer 2 Report

All my comments have been addressed.

Reviewer 3 Report

I thank the authors for their efforts to clarify the manuscript